Mixing state and particle hygroscopicity of organic-dominated aerosols over the Pearl River Delta Region in China

Juan Hong[1,2,3,4], Hanbing Xu[5], Haobo Tan[6*], Changqing Yin[6], Liqing Hao[7], Fei Li[6],
Mingfu Cai[8], Xuejiao Deng[6], Nan Wang[6], Hang Su[1,3], Yafang Cheng[1,3], Lin Wang[4*],
Tuukka Petäjä[2], Veli-Matti Kerminen[2]

[1]Institute for Environmental and Climate Research, Jinan University, Guangzhou, Guangdong 511443, China
[2]Department of Physics, University of Helsinki, P.O. Box 64, Helsinki 00014, Finland
[3]Multiphase Chemistry Department, Max Planck Institute for Chemistry, Mainz 55128, Germany
[4]Shanghai Key Laboratory of Atmospheric Particle Pollution and Prevention, Department of Environmental Science & Engineering, Fudan University, 220 Handan
Road, Shanghai 200433, China
[5]Experimental Teaching Center, Sun Yat-Sen University, Guangzhou 510275, China
[6]Institute of Tropical and Marine Meteorology/Guangdong Provincial Key Laboratory of Regional Numerical Weather Prediction, CMA, Guangzhou 510640, China
[7]Department of Applied Physics, University of Eastern Finland, Kuopio 70211, Finland
[8]School of Atmospheric Sciences, Guangdong Province Key Laboratory for Climate Change and Natural Disaster Studies, and Institute of Earth Climate and Environment System, Sun Yat-sen University, Guangzhou, Guangdong 510275, China

Correspondence to: Haobo Tan (hbtan@grmc.gov.cn) and Lin Wang
(lin_wang@fudan.edu.cn)

Abstract

Simultaneous measurements of aerosol hygroscopicity and particle phase chemical composition were performed at a suburban site over the Pearl River Delta Region in the late summer of 2016 using a self-assembled Hygroscopicity Tandem Differential Mobility Analyzer (HTDMA) and an Aerodyne Quadruple Aerosol Chemical Speciation Monitor (ACSM), respectively. Hygroscopic growth factor (HGF) of Aitken
(30 nm, 60 nm) and accumulation mode (100 nm, 145 nm) particles was obtained under 90% relative humidity (RH). An external mixture was observed for all-sized particles during this study, with a dominant mode of more hygroscopic (MH) particles as aged aerosols dominated due to the anthropogenic influence. The HGF of less hygroscopic (LH) mode particles increased, while their number fractions decreased, during the
daytime due to a reduced degree of external mixing probably from the condensation of gaseous species. These LH mode particles in the early morning or late afternoon could be possibly dominated by carbonaceous material emitted from local automobile exhaust during the rush hours. During polluted days with air masses mainly from the coastal areas, the chemical composition of aerosols had a clear diurnal variation and a strong
correlation with the mean HGF. Closure analysis was carried out between the HTDMA-measured HGF and the ACSM-derived hygroscopicity using various approximations for hygroscopic growth factor of organic compounds (HGF$_{org}$). Considering the assumptions regarding the differences in the mass fraction of each component between PM$_1$ and 145 nm particles, the hygroscopicity-composition closure was achieved using
HGF$_{org}$ of 1.26 for the organic material in the 145 nm particles and a simple linear

relationship between HGF$_{org}$ and the oxidation level inferred from the O:C ratio of the organic material was suggested. Compared with the results from other environments, HGF$_{org}$ obtained from our measurements appeared to be less sensitive to the variation of its oxidation level, which is however similar to the observations in the urban atmosphere of other megacities in China. This finding suggests that the anthropogenic precursors or the photo-oxidation mechanisms might differ significantly between the suburban/urban atmosphere in China and those in other background environments. This may lead to a different characteristics of the oxidation products in secondary organic aerosols (SOA) and therefore to a different relationship between HGForg and their O:C ratio.

## 1. Introduction

Aerosol hygroscopicity describes the interaction between aerosol particles and ambient water molecules at both sub and supersaturated conditions in the atmosphere (Topping et al., 2005; McFiggans et al., 2006; Swietlicki et al., 2008). It is a key property to affect the size distribution of ambient aerosols and can indirectly give information on particle compositions (Swietlicki et al., 2008; Zhang et al., 2011). It also plays an important role in visibility degradation and multiphase chemistry due to an enlarged cross-section area of aerosol particles after taking up water in humid environment (Tang et al., 1996; Malm et al., 2003; Cheng et al., 2008; Liu et al., 2013; Li et al., 2014; Zheng et al., 2015; Cheng et al., 2016). Moreover, it determines the number concentration of cloud condensation nuclei and the lifetime of the clouds, which in turn affect the regional and global climate indirectly (Zhang et al., 2008; Reutter et al., 2009; Su et al., 2010; IPCC, 2013; Rosenfeld et al., 2014; Schmale et al., 2014; Seinfeld et al., 2016; Zieger et al., 2017).

Hygroscopicity measurements have been conducted in numerous laboratory and field conditions around the world. Different observational findings related to hygroscopic properties of particles and their chemical composition were obtained for aerosols from various environmental background conditions. (Bougiatioti et al., 2009; Park et al., 2009; Swietlicki et al., 2008; Asmi et al., 2010; Tritscher et al., 2011; Whitehead et al., 2014; Hong et al., 2015; Chen et al., 2017). Recent studies have specially focused on the hygroscopicity of organic material, as atmospheric aerosols normally contain a large number of organic species, which exhibit highly various water uptake abilities. Previous works have extensively examined and reported the hygroscopicity of the organic fraction in aerosols worldwide, including boreal forest, rural and urban background areas (Chang et al., 2010; Wu et al., 2013; Mei et al., 2013; Hong et al., 2015; Wu et al., 2016). They found that the oxidation level or the oxygenation state of the entire organics, which directly affects their corresponding solubility in water, is the major factor drives the water uptake ability of the organic fraction in aerosols. However, knowledge on the hygroscopicity of organic material and its dependency on the oxidation level of organics in urban background areas under high aerosol mass loading conditions, for instance, in China, where air pollution has become one of the top environmental concerns in recent decays (Chan et al., 2008), is limited.

Due to the fast development of industrialization and urbanization, China has experienced increasingly severe air pollution during the few past decades (Zheng et al., 2015; Wang et al., 2017). High loadings of atmospheric aerosols can reduce visibility and lead to adverse acute and chronic health effects due to penetration and deposition

of submicron particles in the human respiratory system (Dockery et al., 1993; Cabada et al., 2004; Tie et al., 2009). In order to better understand the chemical composition, sources and aging processes of atmospheric aerosols and in turn target the atmospheric pollution problems in China, measurements of atmospheric particles with various properties have increased during the recent years. Hygroscopicity, as an important physico-chemical property of atmospheric particles (Cheng et al., 2008; Gunthe et al., 2011; Cheng et al., 2016), has also been implemented into extensive campaigns in densely populated areas, such as North China Plain (Massling et al., 2009; Liu et al., 2011) and the Yangtze River Delta (Ye et al., 2013). In the Pearl River Delta (PRD) region, a metropolis in southeastern China with high aerosol loadings and low visibility probably due to anthropogenic emissions, hygroscopicity measurements have also been initiated during the past few years (Tan et al., 2013; Jiang et al., 2016; Cai et al., 2017). These previous studies have mainly focused on the statistical analysis of the hygroscopic properties of PRD aerosols and tried to give possible explanations for their temporal variations. However, the relationship between the hygroscopic properties of aerosols in PRD region and the particle phase chemical composition have not yet been systematically constrained, especially the relation of hygroscopic properties of the organic fraction in the particles to its oxidation level. Particularly, a close look at the hygroscopicity and chemical composition of particles during high aerosol loadings is also scarce.

In this study, we measured the size-dependent hygroscopic properties and non-size-resolved chemical composition by a self-assembled Hygroscopicity Tandem Differential Mobility Analyzer (HTDMA) and an Aerodyne Quadruple Aerosol Chemical Speciation Monitor (ACSM) respectively in a suburban site in PRD region. We aim to find the link between the hygroscopicity of aerosols and their chemical composition, with a focus on identifying the hygroscopic properties of the organic material and their O:C dependency for these suburban aerosols. Hygroscopic properties and chemical composition of aerosol particles under high and low aerosol loadings were particularly analyzed separately.

2. Materials and methodology

2.1 Sampling site and air mass origins

The measurements were conducted from 12 September to 19 October 2016 at the CAWNET (Chinese Meteorological Administration Atmospheric Watch Network) station in Panyu, Southern China. The site is located at the top of Dazhengang Mountain, which is in the suburban area of the megacity, Guangzhou. A figure on the geographical location is available in Tan et al. (2013) and Jiang et al. (2016). A detailed description of the CAWNET station and the sampling inlet can be found in Tan et al. (2013) and Cai et al. (2017).

To investigate the relationship between atmospheric aerosol hygroscopicity and the transport paths or source regions of air masses, 72-hour back trajectories of air parcels arriving at CAWNET were calculated at 6-hour intervals using the Hybrid Single-Particle Lagrangian Integrated Trajectory (HYSPLIT) model for this study. The arrival height of the trajectories was chosen to be at 700 m above ground level, which is the mean height of the boundary layer in Guangzhou during the entire experimental period according to the data obtained from European Center for Medium-Range Weather

Forecasts (ERA Interim). Trajectories with similar spatial distributions or patterns were grouped together to generate clusters, representing their mean trajectories and the predominant air mass origins during the campaign.

## 155      2.2      Measurements and data analysis

A self-assembled HTDMA was deployed to measure the hygroscopic growth factor (HGF), mixing state as well as the particle number size distribution (10-1000 nm) of ambient aerosols during this study. A detailed characterization of the HTDMA system and its operating principles are available in Tan et al. (2013b). Briefly, ambient aerosols after passing through a $PM_1$ impactor inlet were first brought through a Nafion dryer (Model PD-70T- 24ss, Perma Pure Inc.) to be dried to RH lower than 10% and were subsequently charged by a neutralizer ($Kr^{85}$, TSI Inc.). These dry particles of four specific mobility diameters ($D_0$; 30, 60, 100 and 145 nm) were selected by the first Differential Mobility Analyzer (DMA1, Model 3081L, TSI Inc.) in the HTDMA system and then were introduced into a membrane permeation humidifier (Model PD-70T-24ss, Perma Pure Inc.) to reach 90% RH. With a second DMA (DMA2, Model 3081L, TSI Inc.) and a condensation particle counter (CPC, Model 3772, TSI Inc.), the growth factor distributions (GFDs) or the mobility diameter ($D_P$) of these conditioned particles at 90% RH and room temperature were measured. The hygroscopic growth factor (HGF, RH=90%) is then defined as:

$$HGF(90\%) = \frac{D_p(RH=90\%)}{D_0}. \tag{1}$$

In practice, growth factor probability density function (GF-PDF) was fitted from the measured GFDs with bimodal lognormal distributions using TDMAfit algorithm (Stolzenburg, 1988; Stolzenburg and McMurry, 2008). After obtaining GF-PDF, the ensemble average hygroscopic growth factor (HGF), number fractions of particles at each mode and the spread of each mode were calculated.

An Aerodyne Quadruple Aerosol Chemical Speciation Monitor (ACSM, Aerodyne Research Inc.) was employed to determine the non-refractory $PM_1$ chemical composition and O:C of submicron aerosol particles with a 50% collection efficiency during the experimental period (Ng et al., 2011). The ratios of oxygen to carbon (O:C) were then estimated by their relationship to the mass fractions of m/z44 (f44) to the total organic mass (Canagaratna et al., 2015). The mass concentration of black carbon was measured by an Aethalometer using a $PM_{2.5}$ inlet (Hansen et al., 1982). Wu et al. (2009) compared the BC concentration in $PM_1$ and $PM_{2.5}$, respectively, and found that BC aerosols mainly exist in the fine particles with roughly 80% of the BC mass in $PM_1$. Due to the limited literature data on BC size distributions in the PRD region, we used this simplified assumption by Wu et al. (2009) to estimate the BC concentration in $PM_1$ for this study. It is necessary to note that the chemical composition of $PM_1$ can be different from those of size-segregated aerosols and the ACSM measures the chemical composition of $PM_1$, which may be significantly different from those of Aitken mode particles. In addition, complimentary measurements for ambient meteorological conditions (e.g. relative humidity, wind direction and wind speed), as well as the particulate matter ($PM_{2.5}$) mass concentration measurements by an Environmental Dust Monitor (EDM, Grimm Model 180) were conducted concurrently during the experimental period.

200         2.3 Closure study

Ambient aerosol particles are mixtures of a vast number and variety of species. In order to estimate the averaged hygroscopicity of ambient particles, the Zdanovskii–Stokes–Robinson (ZSR) mixing rule (Zdanovskii, 1948; Stokes and Robinson, 1966) was
assumed and the hygroscopic growth factor ($HGF_m$) of a mixed particle was calculated by summarizing the volume-weighted HGF of the major chemical components of aerosol particles:

$$HGF_m = (\textstyle\sum_i \varepsilon_i \cdot HGF_i^3)^{1/3}, \tag{2}$$


where $\varepsilon_i$ is the volume fraction of each species and $HGF_i$ is the growth factor of each species present in the mixed particle. The volume fraction of each species was calculated from their individual dry densities and mass fractions from ACSM data (Gysel et al., 2007; Meyer et al., 2009) by neglecting the interactions between different
species. Since ACSM measures the concentration of ions, the molecular composition can be reconstructed from the ion pairing based on the principles of aerosol neutralization and molecular thermodynamics (McMurry et al., 1983; Kortelainen et al., 2017). Several neutral molecules such as $(NH_4)_2SO_4$, $NH_4HSO_4$, $NH_4NO_3$, $H_2SO_4$ and other possible species were therefore obtained. The related properties of each species
necessary for the calculation in Eq. 2 are listed in Table 1. Ensemble values of $HGF_{org}$ were suggested, as the best-fit values of the closure analysis was achieved, which is detailed in Sect. 3.4. As suggested in early studies, the hygroscopicity of organics in the aerosol particles is dependent on their degree of oxygenation inferred from the O:C ratio (Massoli et al., 2010; Duplissy et al., 2011; Hong et al., 2015), hence, we further
estimated $HGF_{org}$ according to the degree of oxygenation presented by the O:C ratio. A similar approach to approximate the hygroscopicity of organics in particle phase based on their O:C ratio is also proposed by Hong et al. (2015). A density value of $1250\,kg/m^3$ was used for the organics to calculate their volume fraction, which was suggested by Yeung et al. (2014) in their closure analysis for aerosols from a similar environment.
        3.   Results and Discussions

            3.1 Overview of measurements

Figure 1 shows the temporal variations of meteorological conditions (e.g., relative humidity, wind direction, average wind speed) and $PM_{2.5}$ as well as BC mass concentration in $PM_1$. In general, RH showed a clear diurnal cycle and a northern wind was frequently experienced during this study. The $PM_{2.5}$ mass concentration varied from 20 to $180\,\mu g/m^3$, with relatively low values (roughly below $50\,\mu g/m^3$) during most
of the time. Previous $PM_{2.5}$ mass concentration measurements at this site have yielded quite similar results (Jiang et al., 2016) at this season. During the period of September 22 to 27, the PRD region experienced stagnant weather conditions, with low wind speeds and fluctuating wind directions near the surface. The stagnant weather leads to the observed increase in the mass concentrations of $PM_{2.5}$ and BC, with up to about two
times higher values compared with the rest of this study.

Figure 2 shows an overview GF-PDF for particles of four different diameters colored with probability density and the mass fractions of the ACSM chemical components as

well as the particle number size distribution (10-400 nm) over the entire measurement period. The white gap in the mass fraction data in the fifth panel is due to an instrument failure. Particles of all sizes showed apparent bimodal growth factor distributions with a mode of more hygroscopic particles and a mode of less hygroscopic particles, indicating the particle population was mainly externally mixed. A similar feature was also observed in the PRD region previously (Eichler et al., 2008; Tan et al., 2013b; Jiang et al., 2016; Cai et al., 2017), as well as in other urban environment around the world (Massling et al., 2005; Fors et al., 2011; Liu et al., 2011; Ye et al., 2013).

In our study, the bimodal distributions had a dominant more hygroscopic (MH) mode for larger particles (100 nm, 145 nm), whereas for smaller particles (30 nm, 60 nm) these number fraction of two modes were approximately of similar magnitude. From the fifth panel in Fig.1, we can see that the total inorganics and organic material had roughly equivalent contributions to the mass fractions in $PM_1$ at the PRD region. This is not a surprise due to the stronger anthropogenic influence in our measurement site. Particle number size distributions below 10 nm were not measured by our setup, so, new particle formation events could not be systematically classified for this study. However, a subsequent particle growth from 10 nm to the accumulation mode was periodically observed. In this study, two distinguished types of days (e.g., one as 'relative clean days' during September 12 to 19 and October 9 to 15 and one as 'polluted days' during September 22, 18:00 to September 27, 9:00) were characterized by their corresponding differences in meteorological conditions, the mass concentration of $PM_{2.5}$ or BC as well as the occurrence of clear particle growth above 10 nm. Distinct analysis of aerosol hygroscopicity, chemical composition as well as air mass origins for these two periods will be further discussed in Sect. 3.3.

### 3.2 Hygroscopicity and mixing state

The diurnal variations of the average HGF of particles of four different sizes are illustrated in Fig. S1. In general, larger particles were more hygroscopic than smaller particles. No strong diurnal pattern of the mean HGF can be concluded from the current results, after taking into account the uncertainties associated with the mean values. This suggests complex sources and aging processes of aerosols at this suburban site.

In the upper panels of Fig. 3, we compared the diurnal variation of the HGFs of particles in the LH and MH mode. HGFs of LH mode of particles of all sizes started to increase after 10:00 am and decrease at about 3:00 pm until reaching their lowest levels at about 8:00 pm. A possible candidate for these LH mode particles could be carbonaceous material emitted from local automobile exhaust during the rush hours, with soot and water-insoluble organics as the major components. These freshly less hygroscopic particles started to age in the atmosphere by condensation of different vapors or multiphase reactions in the daytime, leading to an obvious increase in HGFs of LH mode particles without reaching that of MH mode particles. HGFs of MH mode particles of larger sizes (100 nm, 145 nm) started a slight decrease after about 10 am and then increased again between about the noon and late evening. Particles of this mode are supposed to be more aged than particles in the LH mode, having a substantial fraction of inorganic components such as sulfate and nitrate. However, during daytime when the photochemical activity is stronger, the MH mode particles are expected to experience condensation of different species, especially organics, which are less hygroscopic. Hence, a slightly lower HGF of these particles was observed in the

afternoon than in the morning. In case of smaller particles (30 nm, 60 nm), HGFs of MH group particles appeared to decrease during the afternoon until about 8:00 pm, suggesting that these particles were not long-range transported, but rather secondary formed either locally or from nearby emissions.

The number fractions of different-size particles in each mode are illustrated in the lower panels of Fig. 3. For larger particles (100 nm, 145 nm), MH mode particles dominated over the LH mode particles. For smaller particles (30 nm, 60 nm), the number fraction of LH mode particles decreased dramatically after 12:00 am and increased back to the same level after 6:00 pm. A similar, yet less obvious, pattern was also observed for larger particles. This feature directly suggests that small particles have a lower degree of external mixing during the afternoon compared with the rest of the day, providing further evidence that local traffic emissions may be the major sources of those LH mode particles, especially the ones of smaller sizes.

The hygroscopicity of aerosol particles is ultimately driven by the relative abundances of compounds with different water uptake ability in the particle phase. Hence, we also looked at HGFs of aerosol particles in terms of their direct composition information. Our ACSM measured the non-size resolved chemical composition of particles, which may deviate considerably from that of Aitken mode particles, but be close to that of accumulation mode particles. This requires us to choose HGF of larger particles (100 nm, 145 nm) for the analysis. In Fig. 4, the HGFs of accumulation mode particles correlate quite well with the mass fraction ratio between inorganics and organics + BC. However, the oxidation level of the organic fraction appears to exert only a slight influence on the hygroscopicity of the suburban aerosols, with $R^2$ values of around 0.23. A detailed comparison between the HTDMA-measured HGFs and the predicted HGFs using size-dependent chemical composition will be given below in Sect. 3.4. Gysel et al. (2007) suggested that, compared with HGFs of pure organic particles affected strongly by their oxidation level (Duplissy et al., 2011), HGFs of mixed particles are less sensitive to the properties of uncertainties of growth factor of less hygroscopic compounds in the aerosol phase. This feature might explain why the HGFs of our suburban aerosol were influenced to a lesser extent by the oxidation level of organic compounds than aerosol particles typically studied in smog chamber measurements or measured in a boreal forest environment (Massoli et al., 2010; Tritscher et al., 2011; Hong et al., 2015).

### 3.3 Comparison between polluted and clean days

In order to understand the influence of primary sources and secondary formation to the aerosol loading during different synoptic conditions (e.g., relative clean days and polluted days), we studied the chemical characteristics and physical-chemical properties of aerosols, as well as individual air mass origins, during the two distinguished periods, respectively. Figure 5 shows the diurnal variation of the major species in particle phase during the polluted and relative clean days. Concentrations of all of the displayed species were higher during the polluted period compared with the clean days. This was particularly obvious for $NO_3^-$, whose concentration was almost ten times higher during the polluted days. Wind speeds shown in Fig. 1 were the lowest during the polluted period, enabling local emitted air pollutants such as from traffic and cooking to accumulate. Moreover, a substantial fraction (53%) of the air mass trajectories, shown in Fig. 6, were passing along the coastal areas in the southeast of

China, which is heavily populated. These coastal air masses, together with a considerable fraction (16%) of air masses circulating within the PRD region may potentially transport significant amounts of pollutants, presumably from anthropogenic emissions, to the site. Contrary to this, air masses in the clean days were mainly from the inland areas in the north. These regions are, covered with vegetation and are less influenced by anthropogenic emissions, so air masses coming from there may promote the dilution and clearance of the local pollutants at the observational site.

During the polluted days, $SO_4^{2-}$, $NO_3^-$ and organics had clear diurnal patterns. Concentrations of $SO_4^{2-}$ and organics peaked during the late afternoon, probably due to gas phase condensation or multiphase reactions associated with high levels of $SO_2$ or gaseous organics after long-range transport, as discussed above. Nitrate had higher concentrations in the early morning than in the afternoon. Pathak et al. (2009) suggested that high concentration of particulate nitrate could be explained by the heterogeneous hydrolysis of $N_2O_5$ under high relative humidity conditions. Morino et al. (2006) concluded, using both observation and thermodynamic modeling, that lower temperatures and higher RH cause an enhanced condensation of $HNO_3$ to the particle phase. Fig. S1 shows that RH values were higher in the early morning than other times of the day under polluted conditions. We also looked at gaseous $HNO_3$ concentration, obtained from MARGA measurements and found them to be less than two times higher in polluted conditions compared with clean days. The partition of $HNO_3$ to the particle phase due to condensation might not be able to fully explain the one-order-of-magnitude higher nitrate concentrations in particle phase in polluted days than clean days. Hence, we speculate that the heterogeneous hydrolysis of $N_2O_5$ could be the alternative reason for the production of the observed high concentrations of nitrate in the early morning under polluted condition. During clean days, both inorganic and organic species have lower concentration, with no strong diurnal pattern, which indirectly indicates that the influence of the elevated boundary height on the daily variation of chemical composition was minor. The concentration of BC peaked at around rush hours, suggesting traffic emissions could be one of the major sources of BC.

Considering all examined species together, the difference in the inorganics/organic+BC ratio between early morning and late afternoon was more obvious for the polluted conditions than during the clean days (lower panels of Fig. 5). The averaged O:C ratio during the polluted days was a little bit lower than during the clean days, suggesting that the organic fraction was less oxidized during pollution episode.

The HGFs correlate much better with the contribution of different species to the mass fractions during the polluted days than during the clean days (Fig. 7). However, the oxidation level had a relatively stronger influence on the HGFs during the clean days compared with the polluted days. Taken together, these observations suggest that despite the variability in its oxidation level, the hygroscopicity of the organic aerosol fraction did not vary much during the polluted days.

### 3.4 Hygroscopicity-composition closure

#### 3.4.1   Approximations of the $HGF_{org}$

Hygroscopic growth factors of organic compounds in the ambient aerosols, $HGF_{org}$, cannot be determined from direct observations. However, by conducting closure analysis using different approximation approaches, $HGF_{org}$ was estimated to range widely from about 1.0 to 1.3 for various ambient aerosols in other studies (Gysel et al., 2004; Carrico et al., 2005; Aklilu et al., 2006; Good et al., 2010; Hong et al., 2015; Chen et al., 2017). In this section, we performed a closure study between the measured and predicted HGF using a $PM_1$ bulk chemical composition from the ACSM. An ensemble-mean $HGF_{org}$ (value of 1.1) was determined when the sum of all residuals (RMSE, root mean square error) between the measured growth factors and corresponding ZSR predictions reached a minimum by varying $HGF_{org}$ between 1.0 and 1.3.

By applying this constant $HGF_{org}$, Fig. 8 compares the ACSM-derived HGF with the HTDMA-measured ones for four different-size particles, with the color code indicating the O:C ratio. It is obvious that the degree of agreement increased with increasing particle size. However, the bulk aerosols mainly represent the chemical composition of aerosol particles near the mass median diameter of the mass size distribution of ambient aerosol particles (Wu et al., 2013). The question then arises as to which extent the size-resolved chemical composition of aerosols (for instance, 100 nm and 145 nm particles) is comparable with the one of the bulk aerosol. Previous studies (Cai et al., 2017; Cai et al., 2018) reported that the average organic mass fraction of PM1 were about 25% and 16% lower than those of 100 nm and 145 nm particles respectively measured by High-Resolution AMS (HR-AMS) during the same season of 2014 at the same measurement site. Correspondingly, the average inorganic mass fraction of $PM_1$ were about 25% and 16% higher than those of 100 nm and 145 nm particles obtained in their results. Due to insufficient information of the size-resolved chemical composition of ambient aerosols, we hence made an arbitrary assumption by applying the results from Cai et al. (2017). In this section, we considered the mass fraction of organic being 25% and 16% higher and a corresponding lower inorganic mass fractions (ammonium sulfate mass fraction is decreased) at smaller sizes (100 nm and 145 nm) compared to the bulk aerosol. In addition, we assumed a 20% uncertainty in theses suggested values, thus resulting in 25%±3% and 16±3% of elevation in organic mass fractions in the 100 nm and 145 nm particles for current study. This would lead to larger values of $HGF_{org}$ as 1.23±0.02 (100 nm particles) and 1.26±0.03 (145 nm particles) when assuming different chemical compositions of size-resolved particles compared to the bulk aerosols, see Fig. 9. In contrast to the results from bulk chemical composition, the closure for 100 nm particles considerably improved, as the RMSE value between the HTDMA_HGF and ACSM_HGF decreased from 1.61 to 0.87 with more than 90% of the data were within 10% closure. The closure for 145 nm particles did not show any significant improvement, with no reduction in the RMSE value. However, the newly-determined HGForg is expected to be more accurate than the one reported in the previous section, as assumptions of size-dependent chemical composition was considered even though with some uncertainties. In addition, the newly-obtained $HGF_{org}$ was close to the one  (1.18) by Yeung et al. (2014), who studied the hygroscopicity of ambient aerosols in September 2011 at the HKUST Supersite, less than 120 km away from our measurement site.

Previous studies suggest that a single ensemble $HGF_{org}$ approximation might not be capable of evaluating the hygroscopicity of ambient aerosols from different sources with various characteristics. Hence, the $HGF_{org}$ approximation according to the O:C

ratio was tested using the chemical composition of both bulk aerosols and size-resolved particles based on previous assumptions, respectively. To facilitate our comparison, the closure analysis was only conducted for the 145 nm particles. The relation between $HGF_{org}$ and the O:C ratio based on the chemical composition of bulk aerosols was obtained as follows:

$$HGF_{org} = 0.31 \cdot O{:}C + 0.88. \tag{3}$$

This closure was no better than the one shown in Fig. 8 using a constant $HGF_{org}$, both being based on the chemical composition of bulk aerosols, and there was little change in the RMSE value (from 0.63 to 0.62). By taking into account of the variation of the O:C ratio, $HGF_{org}$ ranged from 0.9 to 1.2 when using Eq. 3 with around 80% of the data having values larger than 1. This finding implies that the approximation in Eq. 3 may introduce huge errors, as 20% of the values of $HGF_{org}$ were not physically correct. The closure considering size-dependent chemical composition of aerosols from previous assumptions is shown in Fig. 10, with a new relation between $HGF_{org}$ and the O:C ratio as:

$$HGF_{org} = (0.32 \pm 0.01) \cdot O{:}C + (1.10 \pm 0.04). \tag{4}$$

The closure was somewhat better than in Fig. 8 according to the slightly lower RMSE value (0.58 vs. 0.63). In addition, $HGF_{org}$ ranged from 1.1 to 1.4 with the varying O:C ratio, and there were no $HGF_{org}$ values smaller than unity, indicating that the new relation in Eq. 4 seems more widely applicable than the one in Eq. 3. In general, by looking at the fitted slopes being much less than unity with consideration of all the discussion above, we are concerned that other potential uncertainties may remain in the closure analysis between the measurements and predictions. This motivates us to make a comprehensive uncertainty analysis of the hygroscopic-composition closure. It is important to note that the uncertainty analysis below is taking into account the aforementioned assumption regarding the size-dependent chemical composition of aerosols.

### 3.4.2    Uncertainties of hygroscopicity-composition closure

Swietlicki et al., (2008) discussed the sources of error associated with HTDMA measurements and concluded that the stability and accuracy of DMA2 RH should be controlled well to maintain the nominal RH (for instance 90%). The accuracy of DMA2 RH in our system was controlled to be 90±1%. This will result in a variability in the measured HGF of ±0.04 around the reported HGF. The bias uncertainty (2.3%) associated with RH accuracy are generally smaller than the estimated uncertainty (10%) reported in HTDMA measurements (Yeung et al., 2014). For hygroscopicity-composition closure, this biased HGF will lead to a change of 2.1% in HGForg with respect to the previously-determined value of 1.26.

Other uncertainties pertain to the densities used for organic materials and black carbon. The density value is estimated to range between 1000 and 1500 kg/m$^3$ for organic materials (Kuwata et al., 2012) and 1000 and 2000 kg/m$^3$ for black carbon (Sloane et al., 1983; Ouimette and Flagan, 1982; Ma et al., 2011). The calculated uncertainty in the ACSM-derived HGF using the density value at each extreme for organic materials and black carbon is less than 3.2% and 2.0%, respectively, both having relatively small effect on the determination of the constant value of $HGF_{org}$.

Another source of uncertainty comes from the measurement of aerosol mass concentration performed by the ACSM and Aethalometer. Bahreini et al. (2009) did a comprehensive uncertainty analysis on aerosol mass concentration measurements using an Aerosol Mass Spectrometer (AMS), having similar operating principle as the ACSM, of which systematic biases are not available. Their study reported an overall uncertainty of 30% for AMS measurements and concluded that it might be better for ground-based studies. Jimenez et al. (2018) gave accuracies of 5-10% from other AMS practitioners and claimed that these estimated accuracies might be too small. Hence, we used an overall uncertainty of 20% for the mass concentration measurements in this study. The uncertainty in the BC measurements given by the manufacture of the Aethalometer is within 5% (Hansen et al., 2005; Zhang et al., 2017). The effect of the perturbation in aerosol mass concentration of each species on the ACSM-derived HGF as well as the determination of $HGF_{org}$ are summarized in Table. 2. The change in the mass concentration of sulfate exerts the largest effect on the ACSM-derived HGF as well as the corresponding HGForg, which is not surprising since sulfate contributes the highest fraction in more hygroscopic component in aerosols.

In general, uncertainties were relatively low for each individual case discussed above. It is possible that the contribution from multiple factors could reduce the overall uncertainties. The greatest uncertainty aforesaid may still arise from the chemical composition of size-segregated aerosols, since the performance of the closure and the approximations of $HGF_{org}$ were most sensitive to changes in the mass concentration of sulfate and organic materials in aerosols. Except for the reasons discussed previously, other factors may also cause potential effects on the hygroscopicity closure. Pajunoja et al. (2015) showed that phase state of organic aerosols, which varies with ambient conditions, might have an effect on the determination of hygroscopicity of organic fraction in aerosols. Previous studies (Suda et al., 2014) suggested that the interaction between inorganic and organic materials within the particle phase might alter the hygroscopicity of organics in mixtures and speculated that ZSR mixing rule may not hold for inorganic dominated aerosols (Hong et al., 2015).

Nevertheless, the interpretation of the hygroscopicity-composition closure and different approximation of $HGF_{org}$ above reveals that in order to estimate accurately the properties of ambient aerosols, we might need to have precise measurements of chemistry, including the size-dependent chemical composition of the aerosols, as well as a better prediction model for HGF.

### 3.4.3    Closure analysis for polluted and clean days

A similar analysis for the hygroscopicity-composition closure similar to that in Sect. 3.4.1 was performed separately for the polluted and clean days. We kept adopting the previous assumption in Sect. 3.4.1 considering the size-dependent chemical composition of aerosols in the current section. The ensemble-mean $HGF_{org}$ value was quite close to each other between the polluted and clean days ($HGF_{org}$=1.30 and 1.28, respectively), and each closure is shown in Fig.S3. These values are similar to the one previously determined ($HGF_{org}$ of 1.26) for the entire experimental period. A good closure was achieved during the polluted days with a substantially high $R^2$ value (0.82), whereas during the clean days, the ACSM-derived HGF did not correlate well with the

one measured by HTDMA, indicating that other factors, such as the O:C ratio of organic material, might have affected the achievement of the closure.

We adopted an O:C dependent hygroscopicity of organic material in the closure analysis separately for the polluted and clean days. The resulting closure is illustrated in Fig. 11. Compared with the clean days, the hygroscopicity of organic material was found to be less dependent on the O:C ratio during the polluted days. This finding is consistent with the previous discussion on Fig.7, stating that the oxidation level had a relatively stronger influence on the HGFs during the clean days compared with the polluted days. This indicates that the organic compound, even with similar hygroscopicity may contain varying chemical species resulting from different sources or atmospheric processes during these two distinct periods. As previously stated in the manuscript, the aerosol particles appeared to have been from long-range transported during the polluted days, having a longer aging history. The organic material in these aerosol particles were fully oxygenated with a similar hygroscopicity, even for different O:C ratios. However, during the clean days, the aerosol particles were mainly from local emissions, or formed locally without complex aging histories. The changes in HGForg revealed the oxidation state of this locally-formed organic material.

### 3.5 Synthetic Comparisons

A number of field studies have examined the relationship between the hygroscopicity of organic compounds and their oxidation level for ambient aerosols from various representative organic aerosol sources (Chang et al., 2010; Chen et al., 2017; Duplissy et al., 2011; Hong et al., 2015a; Mei et al., 2013; Wu et al., 2013, 2016). The empirical relationship obtained from our results and these earlier studies are compared and described below in Fig. 12. It is important to note two aspects before our discussion. First, Eq. 4 considering a size-dependent chemical composition of aerosols is used here for comparison, as it has a wider application than Eq. 3. Secondly, the results from other studies shown in Fig. 12 were obtained using the hygroscopicity parameter ($\kappa_{org}$) (the left y-axis), while in this study we obtained values of HGF$_{org}$. Both parameters represent a quantitative measure of the hygroscopicity of organic material. Hence, we converted our obtained HGForg to hygroscopicity parameter $\kappa$ by the procedure given in Petters and Kreidenweis (2007) and plotted the O:C dependent hygroscopicity parameter $\kappa$ in black line in Fig. 12.

All listed studies show that the hygroscopicity of organic matters generally increases with an increasing organic oxidation level, with significant variance in the fitting slopes among all of the empirical relationships. For aerosols from near remote (Duplissy et al., 2011; Hong et al., 2015) or rural background (Chang et al., 2010) areas, covering little or no influence from anthropogenic activities, the value of O:C exhibits a stronger impact on the water uptake ability of organic materials. This indicates that the oxidation potential from photo-oxidation in the atmosphere of these backgrounds is a critical factor in determining the characteristics of organic materials. Similar to aerosols formed from biogenic precursors, the apparent O:C dependency on the hygroscopicity of organics is obvious for peat burning aerosols (Chen et al., 2017), mostly due to the complexity in the types of biomasses.

In the suburban or urban atmosphere of megacities in China (e.g. Beijing and Guangzhou), the hygroscopicity of organic material was almost constant as shown in

this study and by Wu et al. (2016), being much less sensitive towards the variation in their oxidation level. It is not surprising to observe this similar O:C dependence on hygroscopcity of organic material in the rural background areas of Germany by Wu et al. (2013). This might be explained by the fact that their measurement site is located in central Germany where anthropogenic activities cannot be neglected. Wu et al. (2016) discussed that the addition of either an alcohol or a carboxylic function could both elevate the O:C ratio of the original organic aerosols. However, the corresponding hygroscopicity of these organic products may not be increased to the same extent compared with the increase in the values of O:C. This could be a possible reason to explain that the variation of O:C of organic aerosols is not necessarily responsible for the changes in hygroscopicity. In contrast, $\kappa_{org}$ of aerosols at an urban site in Pasadena, California, in US exhibited a stronger increase with an increasing O:C ratio (Mei et al., 2013). They found that the relationship of their study is in line with that obtained from HTDMA measurements of SOA formed from 1,3,5-trimethylbenzene (TMB), a surrogate for anthropogenic precursors (Duplissy et al., 2011). They also deduced that the major components in SOA from TMB photooxidation are mainly mono-acids, which are quite water soluble. It is also interesting to observe that the results by Lambe et al. (2011) showed quite similar parametrization of HGForg and O:C dependence compared with the one for current study. They used a Potential Aerosol Mass (PAM) flow reactor to study the hygroscopicity of organic aerosols from the oxidation of alkanes and terpenoids, suggesting the precursors of our organic aerosols in this study might have similar properties or same origins as these compounds in their study. The comparisons of $\kappa_{org}$ or $HGF_{org}$ as a function of O:C within these aforementioned studies suggest that anthropogenic precursors or the photo-oxidation mechanisms, might differ significantly between the suburban/urban atmosphere in China and those in the urban background of West US. This may lead to a distinguished characteristics of the oxidation products in SOA and therefore to a different relationship between $\kappa_{org}/HGF_{org}$ and O:C.

## 4. Summary and conclusions

The hygroscopic growth factor distribution obtained in the late summer of 2016 at Panyu CAWNET station in PRD region suggests that this suburban aerosol population with a strong anthropogenic influence was almost always externally mixed. The diurnal variation of the HGF of the LH and MH mode particles of four sizes suggests that the LH mode particles were probably from local emissions, whereas the MH mode particles had a longer aging history. During daytime, an external mixing of particles decreased due to the condensation of different gaseous species onto them, which was particularly obvious for Aitken mode particles. The contribution of different species with various water affinities to the particle composition determines the variation of the mean HGF in general. However, the oxidation level of organics appeared to influence the hygroscopicity of the suburban aerosols only slightly.

The stagnant meteorological conditions favored the accumulation of pollutants originating from coastal areas in the southeast China during the polluted days. During these days, the hygroscopicity of the organic aerosol fraction was estimated to vary little despite the variability of its oxidation level. The atmosphere was cleared by the air masses from the north during clean days.

The ACSM-derived HGF correlated better with the HTDMA-measured ones for larger particles (100, 145 nm particles) compared with smaller particles (30, 60 nm particles). From the closure analysis, considering the assumption of a size-dependent chemical composition of aerosols, a new relation between the hygroscopic growth factor of organic compounds and their oxidation level was obtained for the suburban aerosols

over the PRD region during the experimental periods: $HGF_{org} = (0.32\pm0.01) \times O:C + (1.10\pm0.04)$. Clearly, a moderate hygroscopicity of organic materials, with values of $HGF_{org}$ ranging between 1.1 and 1.3, was observed and it exhibited a weak dependence on the O:C ratio for the current study. Comparison of this relation between polluted and clean days indicate that the organic material even with similar hygroscopicity during

these two distinct periods may contain varying chemical species resulting from different sources or atmospheric processes.

   The PRD region as one of the densely populated areas in China represents a geographical location in Asia under the subtropical marine monsoon climate system.

However, these issues obtained from our results above have been discussed very little earlier, which thereby reflects a general value of our contribution. The comparison with earlier studies regarding the relationship between $HGF_{org}$ and O:C ratio indicates that there are substantial differences, but also some similarities, in the properties of organic compounds in aerosols among different environments, especially in urban areas. This

motivates us to extend our measurement network in the future to understand better the generality of the relationship between the hygroscopicity and the oxygenation of the organic compounds.






Acknowledgements

This work was supported by the National Key Project of the Ministry of Science and Technology of the People's Republic of China (2016YFC0201901, 2016YFC0203305 , 2017YFC0209505) and the Kone-Fudan Nordic Center through Kone Foundation.
This research has also received funding from the National Key Project of MOST (2016YFC0201901), Natural Science Foundation of China (No. 41705099, 41575113, 4160050448 and 91644213) and the Royal Society Newton Advanced Fellowship (NA140106).

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

Table 1. Hygroscopic growth factors of all compounds and their individual density used in the ZSR calculation.

| Compounds | Density (kg m$^{-3}$) | HGF (90%) | |
| --- | --- | --- | --- |
| | | Aitken Mode (30 nm, 60 nm) | Accumulation mode (100 nm, 145 nm) |
| $(NH_4)_2SO_4$[a] | 1769 | 1.66 | 1.70 |
| $NH_4HSO_4$ | 1780 | 1.74 | 1.78 |
| $NH_4NO_3$ | 1720 | 1.74 | 1.80 |
| $H_2SO_4$ | 1830 | 2.02 | 2.05 |
| Organics | 1250[b] | 1.0-1.3[c] | |

a: hygroscopic growth factor and density values of all inorganic was chosen from Gysel et al. (2007)
b: density of organic materials was chosen from Yeung et al. (2014)
c: hygroscopic growth factor for organic materials were varied from 1 to 1.3 according to literature values (Gysel et al., 2004; Carrico et al., 2005; Aklilu et al., 2006; Good et al., 2010; Hong et al., 2015; Chen et al., 2017)

Table 2. Sources of uncertainties associated within hygroscopicity-composition closure, given in terms of three standard deviation and their corresponding contribution to the overall uncertainty in hygroscopicity-composition closure.

| Parameter | Uncertainty (3 standard deviation ) | Uncertainty in measurements | HGForg (relative to 1.26) |
|---|---|---|---|
| RH (DMA2) | 1% | 2.3% in measured HGF | 3.2% |
| Organic density | 18% | 2.6% in ACSM_derived HGF | 3.2% |
| BC density | 33% | 1.0% in ACSM_derived HGF | 2.0% |
| NH4, NO3 mass concentration | 20% | 0.6%, 0.5% | 0.8, 1.6% |
| SO4 mass concentration | 20% | 1.8% | 4.0% |
| Organics mass concentration | 20% | 1.4% | 3.2% |
| BC mass concentration | 5% | 0.1% | 0.8% |







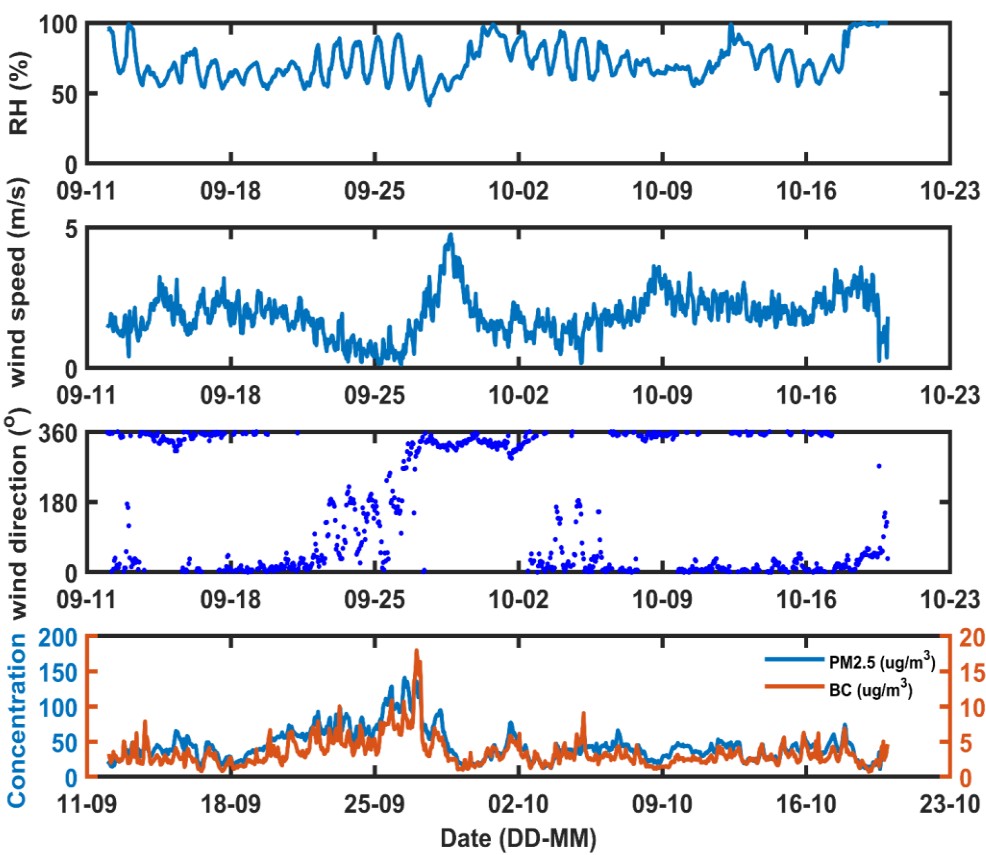

Figure 1. Time series for relative humidity, wind speeds, wind directions and concentrations of PM$_{2.5}$ as well as BC concentration (bottom panel).



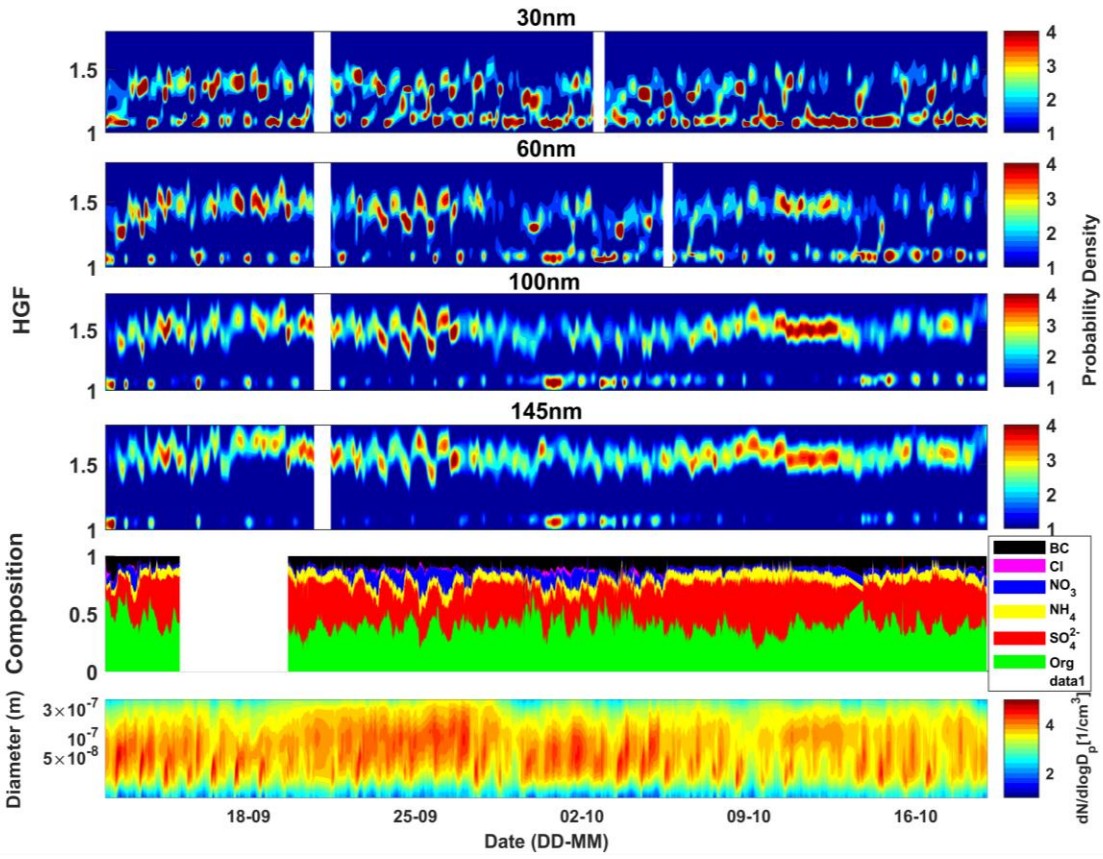

Figure 2. Time series of hygroscopic growth factor distribution for 30, 60, 100 and 145 nm particles using HTDMA in the upper four panels with the color code indicating probability density. Time series of mass fractions of chemical species in submicron particles and particle number size distribution within 10-400 nm using ACSM and DMPS, respectively in the lower two panels.


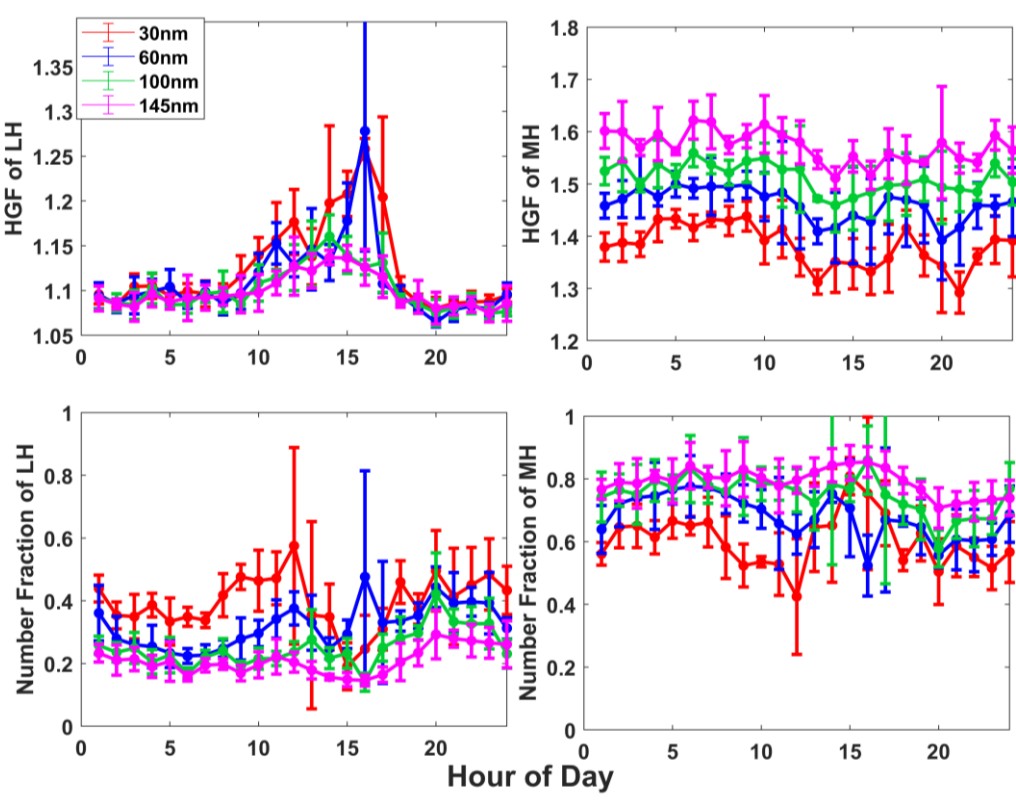

Figure 3. Diurnal variation of the HGF of less hygroscopic (LH) and more hygroscopic (MH) mode particles and their respective number fractions.






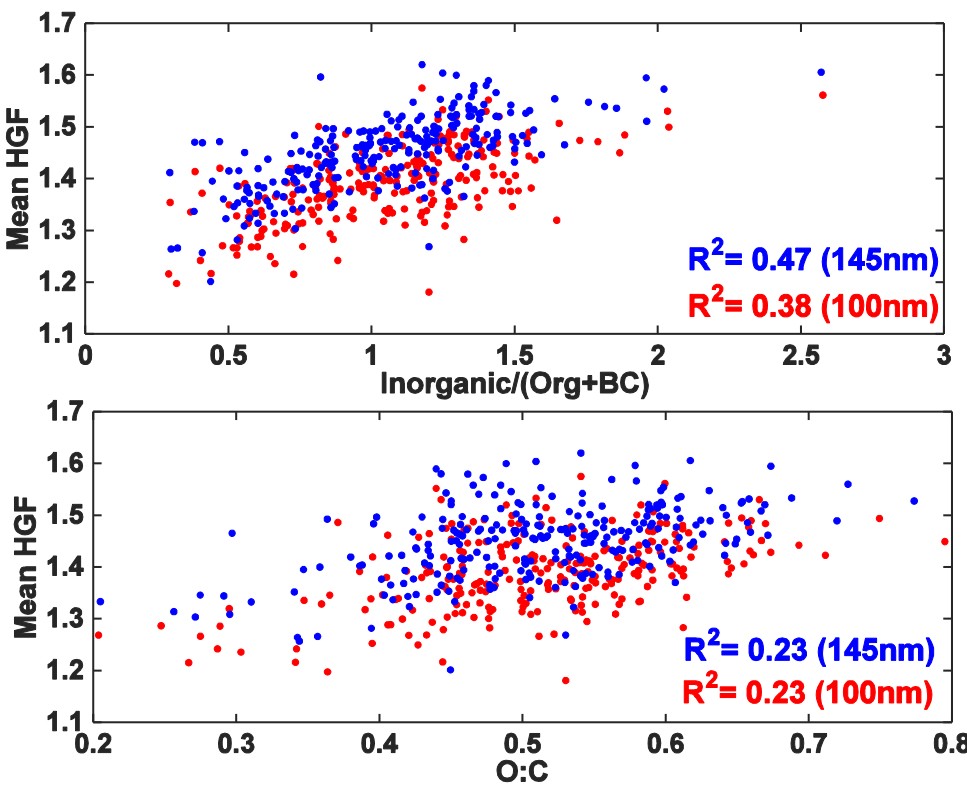

Figure 4. The correlation between the mean HGF of accumulation mode particles (100 nm, 145 nm in size) and the contribution of different species in the particle phase as well as the O:C of the organic materials.

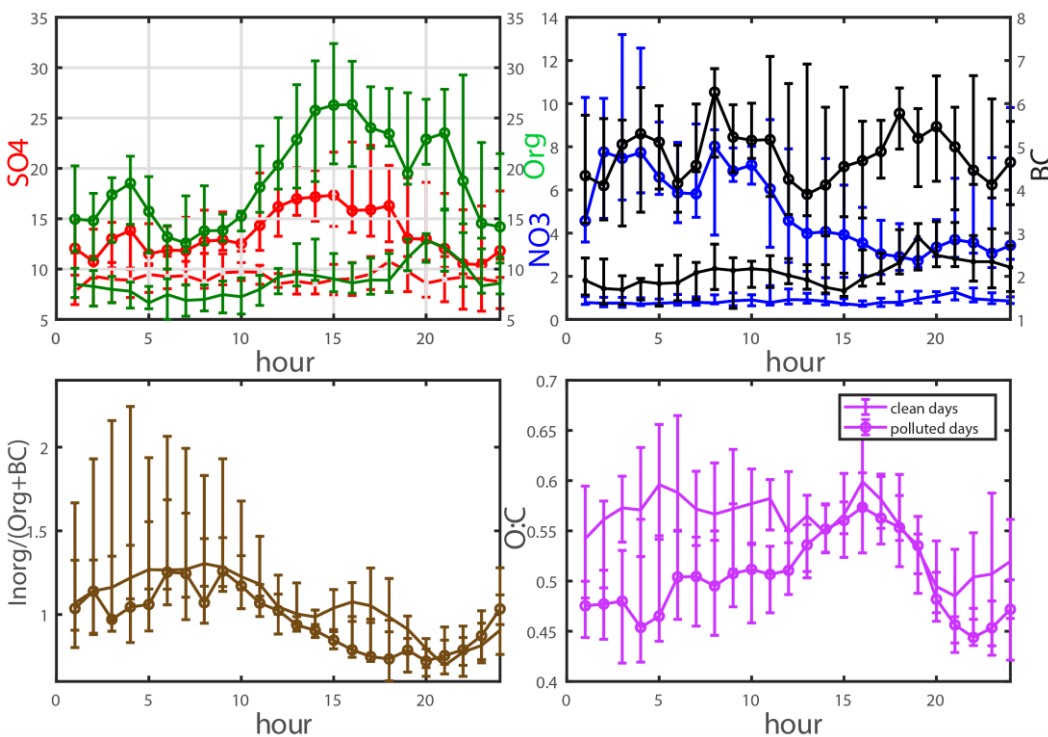

1310      Figure 5. Diurnal variation of mass concentration of $SO_4^{2-}$, organics, $NO_3^-$, BC in particle phase, the O:C ratio of organics and their relative contribution in particle phase composition during clean days and polluted days, respectively.

1315

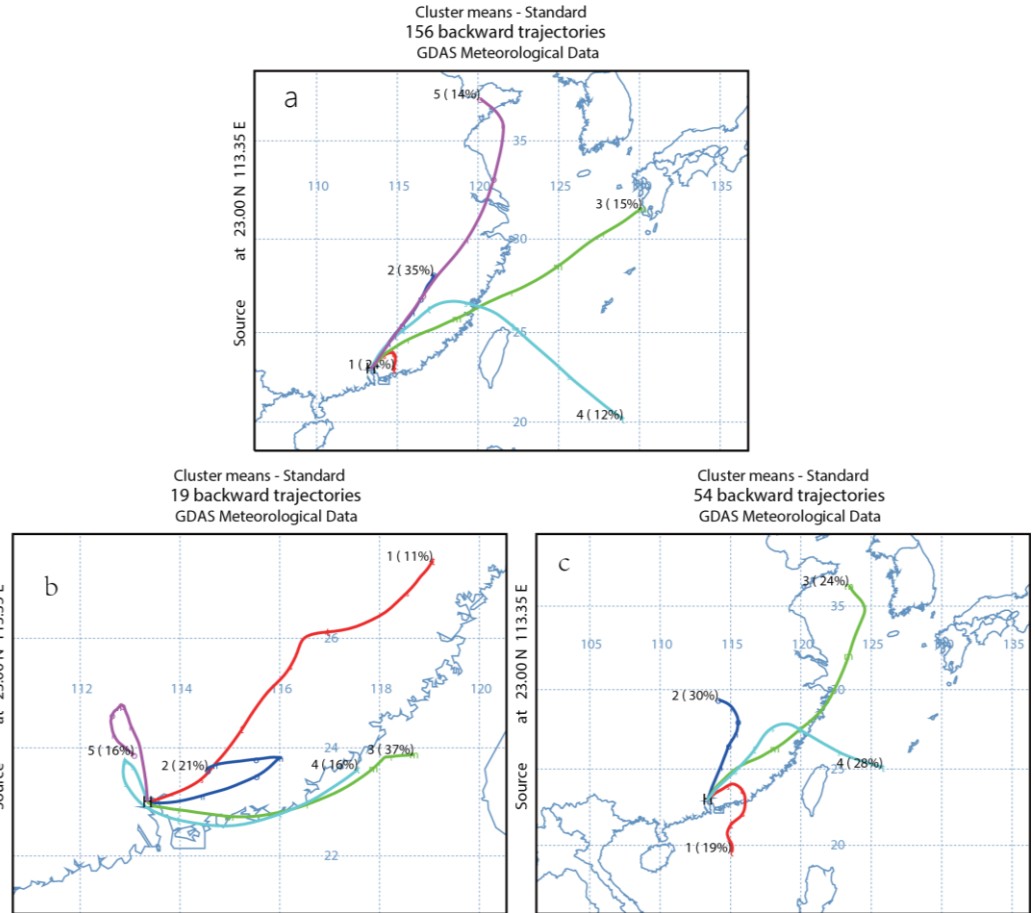

Figure 6. The major clusters for the 72-hour backward trajectory simulation for air masses arriving at the CAWNET Panyu site with an arrival height of 700 m. The upper panel shows the results throughout the whole observational period, while the lower panel on the left side shows the one during polluted days and the one on the right-hand side is for clean days. All trajectories that are near each other were merged to a mean trajectory to represent the entire groups by cluster analysis. The percentage number beside the labeled cluster indicates how many back trajectories can be represented by this cluster.

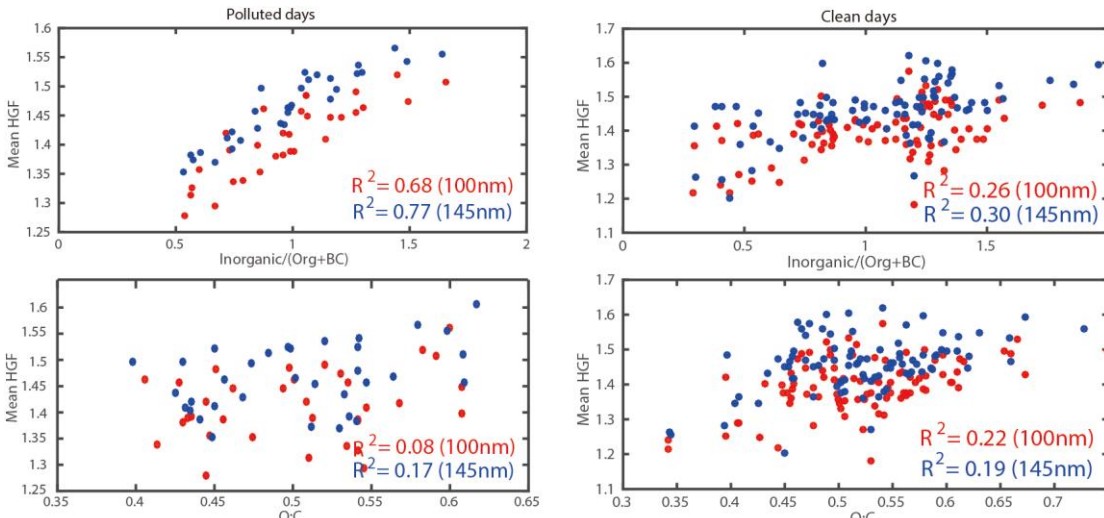

Figure 7. The correlation between the mean HGFs of accumulation mode particles (100 nm, 145 nm in size) and the contribution of different species in the particle phase as well as the O:C of the organic materials during polluted days and clean days, respectively.

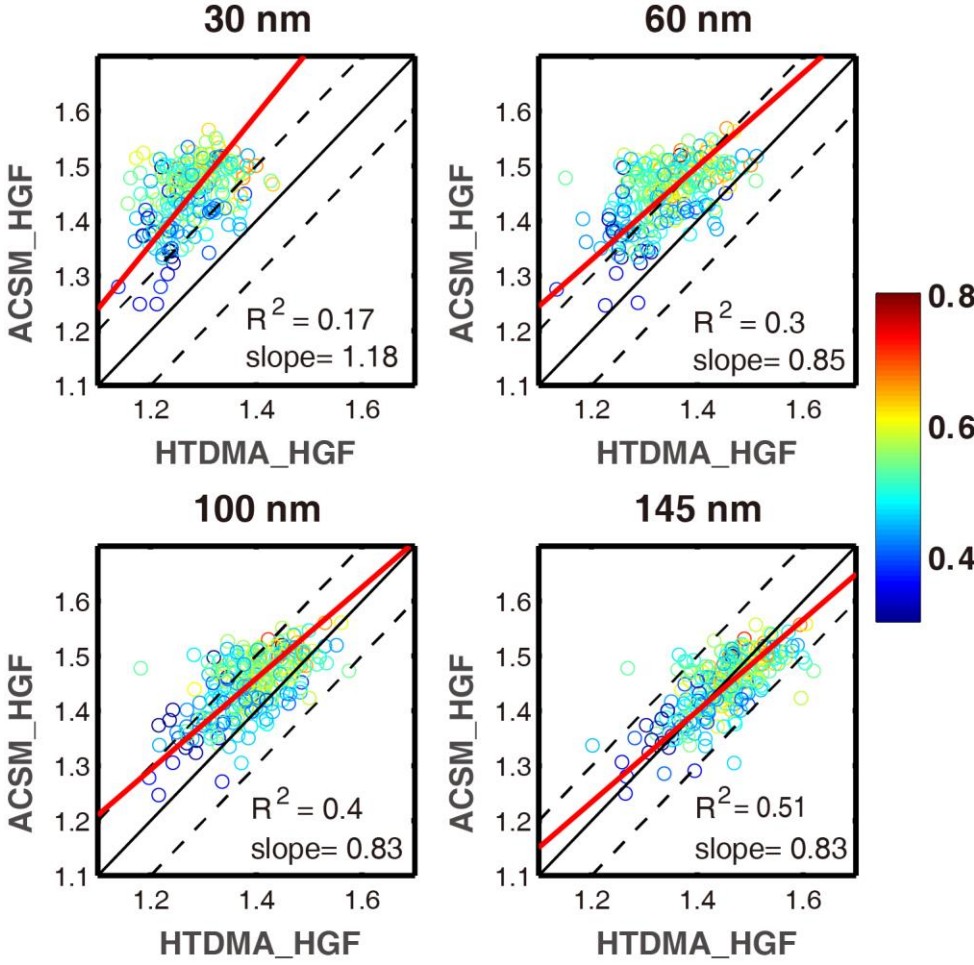

Figure 8. Closure study between the HTDMA-measured HGFs and the ACSM-derived HGFs. The dash lines indicate the 1:1 line, while the red ones are the lines fitted to the data points. The color bar indicates the O:C ratio of the organic aerosol fraction. The black solid lines indicate the 1:1 line and the black dash lines represent ±10% deviation, while the red lines are the lines fitted to the data points. The color bar indicates the O:C ratio of the organic aerosol fraction.

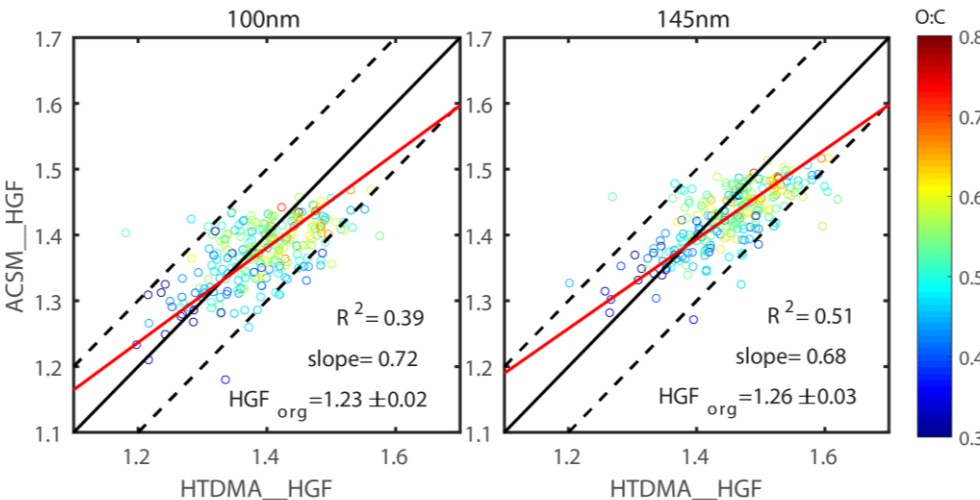

Figure 9. Closure study between the HTDMA-measured HGFs and the ACSM-derived HGFs assuming the average inorganic mass fraction of PM1 were about 25%± 3% and 16% ±3% higher and the average ammonium sulfate mass fraction of $PM_1$ were about 25%± 3% and 16% ±3% lower than those of 100 nm and 145 nm particles. The black solid lines indicate the 1:1 line and the black dash lines represent ±10% deviation, while the red lines are the lines fitted to the data points. The color bar indicates the O:C ratio of the organic aerosol fraction.

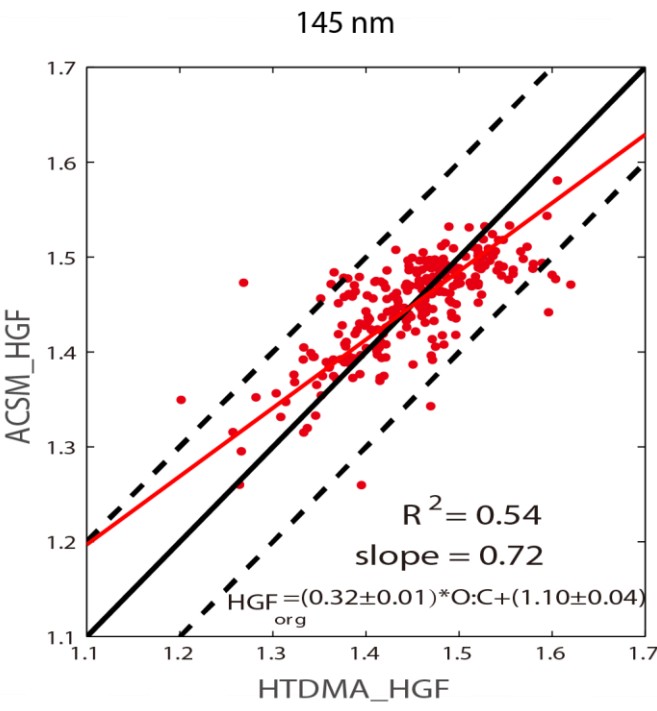

Figure 10. Closure analysis with the best fitting between the measured HGFs and the ACSM-derived ones using the O:C-dependent HGForg for 145 nm particles. The assumption of size-dependent chemical composition of aerosols was considered to determine the ACSM-derived HGFs. The equation is the achieved approximation for $HGF_{org}$ as a function of the O:C ratio of organic aerosol fraction.

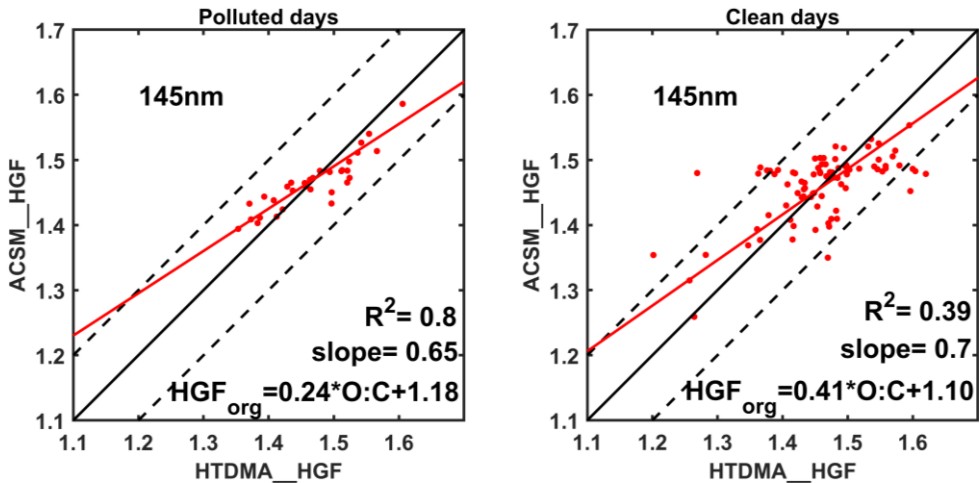

Figure 11. Closure analysis with the best fitting between the measured HGFs and the ACSM-derived ones using the O:C-dependent $HGF_{org}$ for 145 nm particles during the polluted and clean days, respectively. The equation is the achieved approximation for $HGF_{org}$ as a function of the O:C of organic aerosol fraction. During the polluted days, $HGF_{org}$ is less sensitive to the changes in the O:C ratio of organic material compared with the ones during the clean days, indicating different organic species during these two distinct periods.

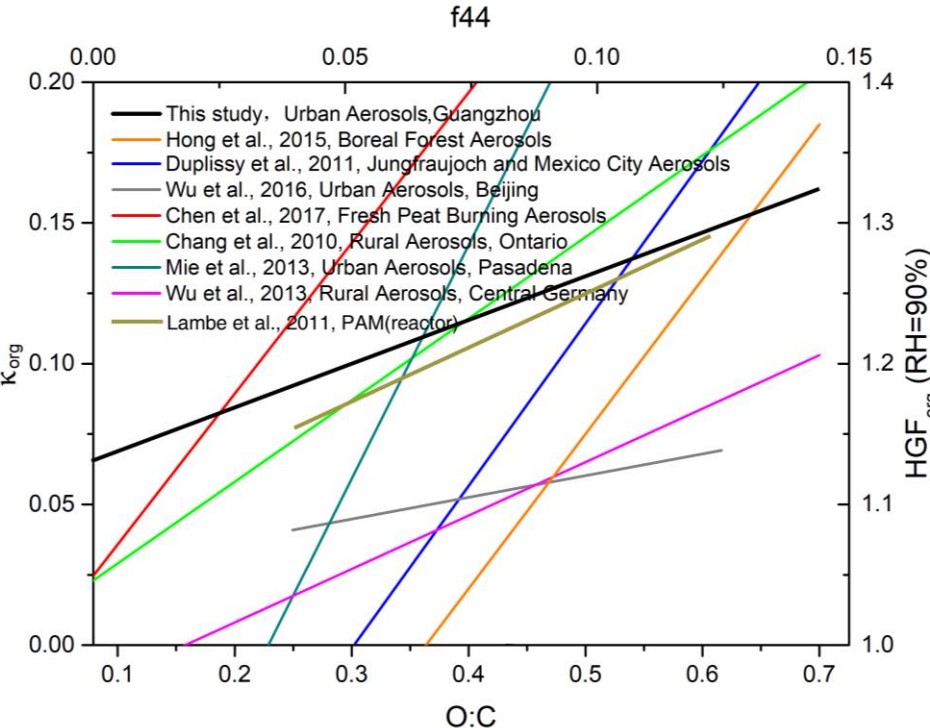

Figure 12. Comparison with earlier studies on the hygroscopicity of organic material with atomic O:C ratio (or *f44* from chemical composition data) obtained from different environmental background areas. In this figure, HGF$_{org}$ of this study was converted to $\kappa_{org}$ for comparison.

Supplement:

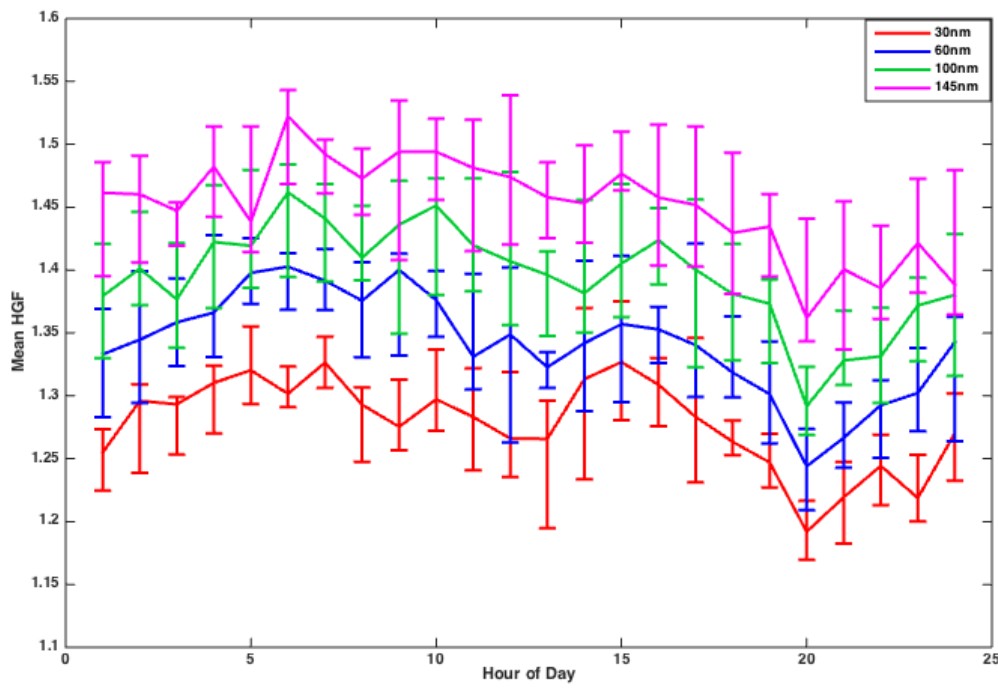


Figure S1. Diurnal variation of the mean hygroscopic growth factor of 30, 60, 100 and 145 nm particles during this study.


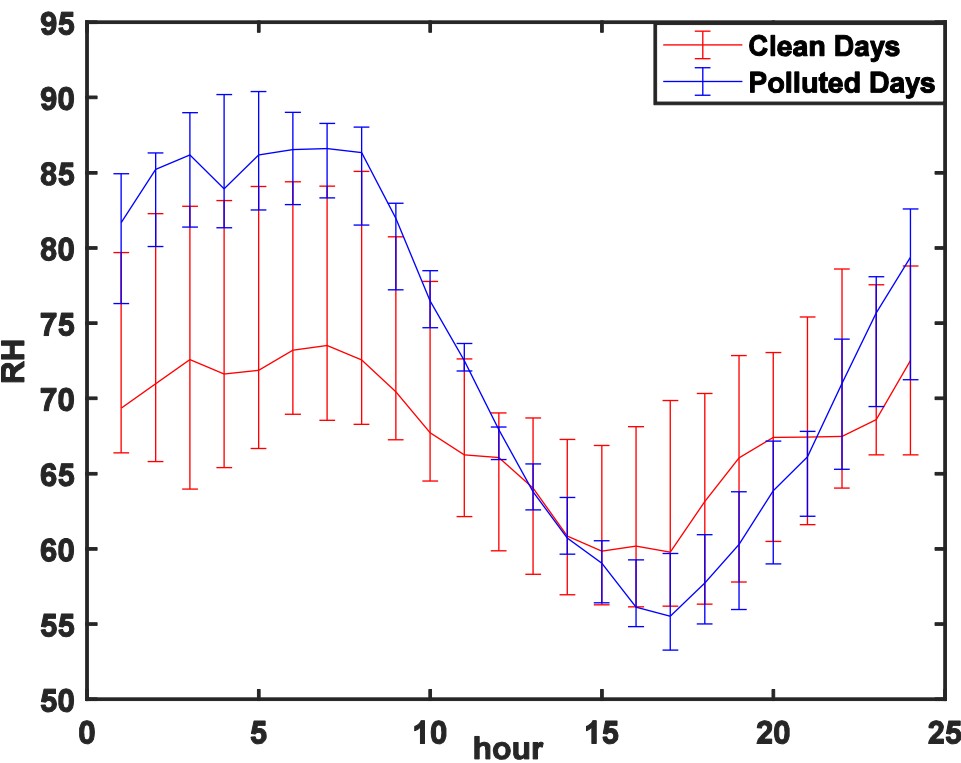

Figure S2. Diurnal variation of relative humidity during the polluted and clean days.





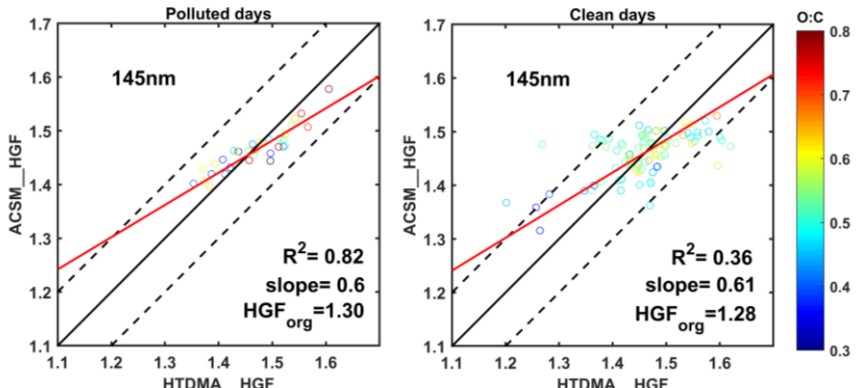

Figure S3. Closure analysis with the best fitting between the measured HGFs and the ACSM-derived ones using constant $HGF_{org}$ for 145 nm particles during the polluted and clean days, respectively.
