# Peer review of "Mixing state and particle hygroscopicity of organic-dominated aerosols over the Pearl River Delta Region in China"

_Atmospheric Chemistry and Physics, 2018_

## Referee Comment (RC1) · Anonymous Referee #1 · 17 Mar 2018

The MS mainly deals with evaluating and interpreting the results of simultaneous measurements of hygroscopicity and chemical composition of aerosol particles in a diameter range of 30–145 nm in the Pearl River Delta Region in late summer. The performed work is definitely of scientific importance and relevance, its findings represent a valuable contribution to the increasing knowledge in this field. Nevertheless, the MS was prepared not enough in a careful way, and should be improved substantially before the paper can be accepted for publication. Some examples of this are given in Minor comments, and some other similar changes are also necessary.

Major comments

1. The authors should describe the exact calculation procedure possibly with mathematical equations for lines 171–174. 2. Sect. 2.1 and later. The arrival height of

150 m above the ground is unusually low, the model performs better at higher levels. This raises the question about the representativity of these air mass trajectories. The authors should discuss and explain their selection, and/or present similar trajectories for at least 2 more but larger arrival heights (e.g. one just below and one above the fee troposphere). In addition to that, the authors should describe how the major trajectory clusters and their frequency in Fig. 7 were exactly derived from the individual trajectories. 3. Aerosol particles are usually grouped as: nearly hydrophobic ($\kappa$<0.10), less hygroscopic ($\kappa$=0.10–0.20) and more hygroscopic ($\kappa$>0.20; Liu et al., Atmos. Chem. Phys., 11, 3479–3494, 2011). The authors may want to follow this attitude, or explain and argue for their own classification. 4. The authors are requested to extend the MS with explicit discussions and estimations of the uncertainty of their major conclusions to prove their significance. In addition to that, smaller changes throughout the MS, e.g. showing standard deviations of slopes for correlation lines in Fig. 10, and similar amendments could also be adopted.

Minor comments

1. It is disturbing that the terms aerosol – aerosols – aerosol particles are not used in a consistent manner in the MS (e.g. lines 64, 123 and 196). The authors should decide which option to use, and should adopt it in a coherent nomenclature. 2. Abbreviation PM stands for particulate matter (as correctly stated in line 191), but it is sometimes used for particle matter mass (e.g. line 30). The explanation of the abbreviation should be given at its first occurrence, and it is redundant to repeated it e.g. in lines 232–233. Furthermore, simply write for instance: "The PM2.5 mass concentration varied" instead of "The range of particle mass concentration (PM2.5) varied". In addition, it is not the range that varies. 3. Hygroscopicity usually refers only to sub-saturated conditions. Clarify line 65, or give appropriate references to back your statement. 4. Micron (e.g. lines 98, 177, Fig. 2) is not an SI unit. Micrometer should be used instead. 5. Remove the repetitions in lines 102–103 considering lines 67–69. 6. Revisit "self-assembly" (line 120) and "self-assembled" (line 152), or use

perhaps laboratory made instead. 7. Clarify lines 185–189. 8. Abscissa of Figs. 1 and 2 shows rather Date than Time, and its format of e.g. "(dd-MM)" could also be indicated. Furthermore, they could have (this) consistent format in both figures. In Fig. 2, how can be the probability in colour coding larger than 1? Explain or modify this. Put word space between measured values and their units everywhere in the figures. Extend the interpretation of your finding and discussions related to Fig. 2 within the frame of the conjunct conclusions of Cheung et al., Atmos. Chem. Phys., 16, 8431–8446, 2016. 9. Figures 3, 4 and 6 show the mean diurnal variations; the label of abscissa should be consistent in three figures; remove the tick label at 25; the time unit as "(HH)" should also be indicated; "during this study" is redundant; avoid abbreviations in the figure captions everywhere. Extend the interpretation of your finding and discussions related to the diurnal plots (Figs. 3 and 4 or Sects. 3.2 and 3.3) within the frame of the similar recent data of Enroth et al., Atmos. Chem. Phys. Discuss., https://doi.org/10.5194/acp-2017-926, 2017. 10. All correlation scatter plots should have squared layout to facilitate better their interpretations. 11. Rounding off strategy should be revised throughout the MS; e.g. of $R^2$ in the figures or HGForg in lines 399, 405 ("1 and 1.3"), 515 or 546. 12. Lines 288–289: Remove "black carbon or" from "with black carbon or soot". 13. Line 319: Consider writing "compounds with different water uptake ability" instead of "compounds of different water uptake ability".

Please also note the supplement to this comment:
https://www.atmos-chem-phys-discuss.net/acp-2018-77/acp-2018-77-RC1-supplement.pdf
* * *

---

## Referee Comment (RC2) · Anonymous Referee #2 · 31 Mar 2018

The manuscript by Hong et al. presents aerosol hygroscopicity and chemical composition measured near a megacity in south China. Size-resolved hygroscopic growth factors (HGF) and PM1 chemical composition were measured using a Hygroscopic Tandem Mobility Analyzer (HTDMA) and an Aerosol Chemical Speciation Monitor (ACSM), respectively. Based on the closure analysis of the two independent measurements, the authors found that the organic particulate material was primarily hydrophobic during the pollution episode. More specifically, the authors derived a new empirical relation between the HGF for organic material (HGForg) and the O:C elemental ratio. The HGForg values predicted by this new relation appear to be significantly lower than that reported in previous literature. Although I find the new dataset is interesting, the key finding of hydrophobic anthropogenic organic particulate matter seems to poorly sup-

ported by the analysis, and can be misleading. One major concern is that the HGForg values derived from the closure analysis are highly uncertain, and the uncertainty is not analyzed in the manuscript. I therefore do not recommend publication in ACP.

Major issues:

1. The closure analysis compares ACSM-derived HGF for PM1 with HTDMA-measured HGF for 100 nm or 150 nm particles. What was the mode diameter (in terms of mass-diameter distribution) of the particle population of PM1? If the mode diameter of PM1 is significantly larger than 150 nm, the estimated HGForg can be significantly under-estimated, because the hygroscopicity generally increases with the diameter in the accumulation mode (at least for 50 nm - 250 nm particles as shown in Liu et al. ACP, 2011). Is it possible that the estimated low HGForg is affected by such systematic biases?

2. In line 328-331, the authors mentioned that the measured HGF values for the mixed particles are less sensitive to the relationship of HGForg vs. O:C. I agree with this statement. However, this implies that the derived parameterization of HGForg = 0.3*O:C + 0.87 can be highly uncertain, as it is not well constrained by the measurements. I noticed that including the O:C-dependent HGForg only improves $R^2$ from 0.5 to 0.51 (Fig. 10 vs. Fig. 9 d). In this case, what is the error associated with the parameterization?

3. Related to the previous comment, the proposed relationship would predict a HGForg value smaller than 1 for O:C < 0.43. This can be misleading. What does a HGForg value smaller than 1 mean? Is it just because of measurement errors?

4. The authors reported a best-fit HGForg value of 1.1. The error bar associated with this value should also be reported. Errors of both HTDMA and ACSM, along with the uncertainties associated bulk vs size-resolved closure analysis should be considered.

5. Even if the HGForg = 1.1 value is accurate, I don't think the authors can assert that "the CCN concentrations would probably be over-predicted as the organic material in

these locations appeared to be close to hydrophobic". Several studies have shown that the CCN activity can be higher than that predicted based on the HGForg at ∼90% RH. For example, alpha-pinene-derived SOA has a kappa value of ∼0.04 at 90% RH, but the CCN-derived kappa value is around 0.1 (see Pajunoja et al., 2015), which is close to the value used in climate models. The low hygroscopicity derived at subsaturation regime does not necessarily indicate a low CCN activity.

Technical issues:

1. The authors should describe how the HTDMA was calibrated and how the data inversion was performed. Did the measurements reproduce literature values for pure compounds (e.g., ammonium sulfate)? Was the RH sensor calibrated? Did the two DMAs have a shift in diameter? Was the DMA transfer function considered in data inversion?

2. The O:C ratio was estimated from the f44 measured by a quadrupole ACSM, presumably less accurate than that measured by the HR-ToF-AMS. What was the uncertainty of the ACSM-derived O:C?

3. Figure 2. Unit of the color bar is missing.

---

## Referee Comment (RC3) · Anonymous Referee #3 · 2 Apr 2018

The authors present a study of low hygroscopicity of organic material in anthropogenic aerosols under pollution episode in China. The data set is rich, but the manuscript has two major deficiencies that should be addressed prior to considering further review. (1) The upper particle size ranges detected by different instruments are quite different: 145 nm for HTDMA, 1um for ACSM, and 2.5 um for Aethalometer. Before drawing any conclusions, the authors should consider the uncertainties caused by different size ranges of particles being measured when integrating all dataset. (2) Most of previous studies showed that the hygroscopic growth factors of secondary organic aerosols are below 1.2. For example, the water soluble organic carbon measured by Martin Gysel et al., has a hygroscopic growth factor of up to 1.17. In this study, the HGForg below 1.1 is actually comparable to those in other observations. Therefore, the low hygroscopicity

of organic material can not be a compelling result. Reference Gysel, M., et al., Hygroscopic properties of water-soluble matter and humic-like organics in atmospheric fine aerosol. Atmos. Chem. Phys., 2004. 4(1): p. 35-50.
* * *

---

## Referee Comment (RC4) · Anonymous Referee #4 · 2 Apr 2018

This manuscript describes measurements of the hygroscopicity of ambient aerosol measured at a surburban site in the Pearl River Delta area of China. Using a Hygroscopicity Tandem Differential Mobility Analyzer (HTDMA), the hygroscopic growth factor (HGF) for 4 aerosol sizes was measured during a 5-week intensive observation period, while an Aerosol Chemical Speciation Monitor (ACSM) measured the chemical composition of the same aerosol population. This observation period included some clear and some polluted days, with air masses that were determined to come from congested and non-congested areas of southeastern Asia. The authors observe a bimodal distribution of HGF for all aerosol sizes, which they attribute to two distinct aerosol populations, one fresh and one aged. They examine the dependence of the HGF on factors such as aerosol mass loading and O:C ratio, and determine that the HGFs observed

here have a much weaker dependence on the O:C ratio than has been previously observed, and suggest that this may be related to the different chemical composition of the local emissions.

This is an interesting paper, and a good data set, and may be publishable in ACP, but I believe the authors need to address several major concerns prior to publication.

General issues:

This paper is attempting to compare many components of the measured aerosol: four different aerosol sizes, two different HGF modes, clean vs polluted conditions, and diurnal trends. It's a complicated set of comparisons, and different sections of the paper address different things. The reader would benefit if the authors would more clearly state what each section is comparing, and only include the most relevant comparisons. For example, the first paragraph of section 3.2 discusses the diurnal trends of the mean HGF, but quickly states that there are no significant trends – probably because in the next section we can see that the LH and MH modes have opposite trends, and the mean HGF, which is the average of the two, sees these trends cancel out. So why include the mean HGF at all? Another example is at the end of section 3.3, where the authors demonstrate that the HGF dependence is different in clean and polluted conditions. If this is true, the authors should be careful in the rest of the paper to distinguish between clean and polluted conditions in their other analyses.

Secondly, the authors should identify the primary message of the manuscript and more clearly describe this result. Is it that the hygroscopicity's lower-than-expected dependence on O:C is attributed to a higher concentration of organics with larger molecular weights? If so, the authors should discuss this further. Are there experimental measurements available to support this? If this is the main conclusion, what should the reader learn from the extensive look into the dependence on inorganics, on the diurnal averages, which is what the majority of the results section is about?

Specific issues:

Line 86 - 90: What is the relevant difference here between oxidation level and the oxygenation state? Insert a sentence here detailing why oxidation level is theoretically correlated with water uptake, since this is an important part of the results.

Line 90 - 92: It is stated that the knowledge of the dependency of hygroscopicity on oxidation level is unknown in urban China. Since this is the main focus of the paper, include a line indicating why this environment is different.

Line 152 - 173: More details about the HTDMA should be included in this section. The second DMA is operating in SMPS mode? How fast/frequent are the scans and therefore what is the time-resolution for retrieval of the HGF? How frequently does the first DMA cycle between the 4 diameter set points? How are doubly- and triply-charged particles that are transmitted by the first DMA handled? Are the particle size distributions plotted in the bottom frame of Figure 2 from SMPS scans by the first DMA or from some other technique?

Line 178: What are 'Ambient-improved' ratios? Either define this term or leave it out and direct the reader to a reference.

Line 184 - 186: Briefly state what the simplified approach is. Is all the BC assumed to be in PM1? Or a weighted fraction?

Line 188: The line "individual size bins" is confusing. I assume the authors are referring to the 4 sizes selected by the first DMA? Replace with something similar to "the ACSM measures only accumulation mode aerosol, and therefore the Aitken mode particles may have a different chemical composition".

Line 191: Briefly state what instrument was measuring the PM2.5 chemical concentrations. An AMS?

Line 209 - 212: What is the justification for assuming the aerosol is completely neutralized? What would the effect be on the results be if it were not completely neutralized?

Line 272 - 282: See comment in General Comments. The paragraph is perhaps unnecessary. What can be learned from looking at the diurnal profile of the mean HGF that isn't learned from looking at the MH and LH components separately?

Line 296 - 299: What is the justification for the assertion that the MH mode particle experience a decrease in HGF during the day because they are uptaking less photore-active species. Do typical reaction rates or back-of-the-envelope calculations support this assertion? Which species are involved? If this is true, how do the authors reconcile the fact the O:C ratio sharply increases during the day, and this paper indicates that there is at least a somewhat positive correlation between O:C and HGF?

Line 305: The authors state here that Hong 2015 and Cai 2017 report that the boundary layer height has an effect on aerosol populations, but later on line 378, they suggest it doesn't. This disagreement should be addressed more fully.

Line 323: The authors state that they can only compare HGFs from the HTDMA and ACSM for larger particles. But they have also demonstrated that larger and smaller particles behave differently. The authors should address any hypotheses for how HTDMA and ACSM might agree for smaller particles.

Line 325: State why HGF is expected to positively correlate with the inorganics/(organics + BC) ratio.

Line 349 and 352: The authors state the percentages 64% and 21% in reference to the back trajectories without discussing where these numbers come from. Furthermore, more information about the trajectories would be helpful, such as error bars on those percentages.

Line 354: Is there an observed increase in ACSM organics on days when the trajectories indicate air masses are arriving from the inland areas? If not, why is that?

Line 390: Do the authors have a suggestion for why this trend (HGF depends on O:C more during clean days than polluted days) is observed? It seems like an important result, yet isn't discussed extensively in the conclusions. Additionally, why is the parameterization of the HGF-to-O:C relationship not done separately for clear vs polluted days?

Line 401: Is there an operational definition for suburban aerosol? Does this just mean an aerosol population that is somewhere between typical urban and rural characteristics?

Line 402 - 405: More detail about the residual fit should be added here. Is the ZSR prediction compared to all the HTDMA measured HGF? Of all sizes? Or just the polluted or clear days? Are different values derived depending on the subset of measured data to compare to?

Line 415: Why was the ACSM not measuring size-selected aerosol in this study, as was done in Yeung et al?

Line 425: More information should be included about how this parameterization was derived. What parameters were allowed to vary, and what was the parameter that was minimized? Is a R2 of 0.51 significantly better than 0.5? In the next paragraph, an improved parameterization is introduced by allowing SOA density to vary. Which parameterization is better? Why does the conclusion section only mention this first parameterization?

Line 430: What is the justification for stating that the hygroscopicity of organics isn't affected by the presence of inorganics?

Line 444: How are the authors accounting for error here? Presumably there is error in the measurement, which propagates through to the derivation of the parameterization.

Line 490: Have the authors plotted the HGF vs the concentration of certain inorganics? Say, vs ammonium sulfate or sulfuric acid to see if there is a larger trend for compounds known to be more hygroscopic?

Figure 2: Remove the dates from under each frame and just put them under the bottom frame. Color bar for the top four frames should be labeled. Additionally, it seems as

though the MH and LH modes both have diurnal cycles between <1 and > 4. If this is simply because the total number of particles has a diurnal profile, it would be easier to see this if it was normalized to the total number of particles. In the bottom plot, because there is only one point on the y-axis, it's hard to see that it's in log space. The boundaries (i.e. 10 -1000 nm) should be indicated, with ticks to show that it is logarithmic.

Figure 3: See Comment on line 272. It's possible that this figure is not needed.

Figure 4: Is this separately out for polluted or clear days? Why not?

Figure 5: What happens if these plots are made with MH or LH HGF instead of the mean?

Figure 7: The colors for these trajectories should be labeled more clearly, and described more fully in the caption and also in the manuscript. Do they represent one representative trajectory? Or a weighted average? What was the spread on those individual trajectories?

Grammatical/Minor:

Line 102: What does "purposes" mean here? Do you mean "properties"?

Line 107: PRD, not RPD

Line 155: Tan et al. 2013b doesn't appear to be in the listed references. Neither is Tan 2013a

Line 159: Why denote the dry mobility diameter as D0? Why not "Dp (0% RH)"?

---

## Referee Comment (RC5) · Anonymous Referee #5 · 9 Apr 2018

General comments

This manuscript presents results of simultaneous measurements of aerosol hygroscopicity and chemical composition in suburban site in Southern China. The measurement period covers almost 1 month and both polluted and relatively clean conditions were observed. I admit that such measurement can be very costly and labor intensive, and therefore the comprehensive set of data presented in the manuscript may carry certain value for the scientific community. However, with the current state of the manuscript, authors' main conclusion is very difficult to sink in for the readers. Authors seem to persist on determining the HGForg and large part of the manuscript is dedicated for that. However, in my opinion, it is obvious from the results that the oxidation level of organics does not affect the hygroscopicity of the suburban aerosols very much, and

that might pretty much be the end of the story for HGFrog. Instead, I would like to see much more in-depth discussion on diurnal variations of LH mode in smaller particles (30 and 60 nm) and how the new particle formation and subsequent growth affects the aerosol hygroscopicity. I therefore recommend that the manuscript may be acceptable for publication in ACP after major restructuring.

Specific comments

152: It is not clear from the manuscript how the HTDMA was operated to obtain the particle number size distribution (10-1000 nm) simultaneously while the instrument was measuring HGF in 4 size classes.

175-178: It is critical to indicate the calibration procedure of ACSM and what calibration parameters were used (e.g. relative ionization efficiency of SO4). Such calibration parameters can critically affect the inorganic and organic mass fractions (and therefore the ensemble HGForg of 1.1).

300-303: The logical basis to support the following conclusion is not clear. "In case of smaller particles (30 nm, 60 nm), HGFs of MH group particles appeared to decrease during the afternoon until about 8:00 pm, suggesting that these particles were not long-range transported, but rather secondary formed either locally or from nearby emissions."

462-519: Extra caution must be taken when comparing k based on supersaturation conditions and HGF based on sub-saturated conditions. The k derived from sub- and supersaturated conditions can be quite different in some cases. In such case, the discussion on potential bias on CCN concentration may not be relevant.

Technical corrections

120: "self-assembly" should appear "self-assembled"

188: what does it mean by "individual size bins"?

[Figure]

330: rephrase "uncertainties of in growth factor"

423: "as followed" should appear "as follows"

[Figure]

---

## Author Comment (AC1) · 9 Jul 2018

Answers to Referee #1

The authors appreciate the time the reviewer has spent on our manuscript, assisting us to produce a higher quality, understandable publication. All the requested comments and suggestions are addressed and introduced to the revised version of the manuscript.

Major comments:

1. The authors should describe the exact calculation procedure possibly with mathematical equations for lines 171–174.

Reply: After obtaining GF-PDF, which was represented as $c(HGF)$, ensemble mean hygroscopic growth factor (HGF), number fraction of particles at each mode and spread of each mode was calculated according to the following equations (Gysel et al., 2009; Tan et al., 2013):

$$HGF_{mean} = \int_0^\infty HGF \cdot c(HGF)d(HGF). \tag{1}$$

$$\sigma = \left(\int_0^\infty (HGF - HGF_{mean})^2 \, c(HGF)d(HGF)\right)^{1/2}. \tag{2}$$

$$NF^{a,b} = \int_a^b c(HGF)d(HGF). \tag{3}$$

Here $HGF_{mean}$ is the number weighted mean hygroscopic growth factor, $\sigma$ is the standard deviation of GF-PDF, which is used as a measure for the degree of mixing of aerosol population. $NF^{a,b}$ represents the number fraction of particles in which $a<HGF<b$.

2. Sect. 2.1 and later. The arrival height of 150 m above the ground is unusually low; the model performs better at higher levels. This raises the question about the representatively of these air mass trajectories. The authors should discuss and explain their selection, and/or present similar trajectories for at least 2 more but larger arrival heights (e.g. one just below and one above the fee troposphere). In addition to that, the authors should describe how the major trajectory clusters and their frequency in Fig. 7 were exactly derived from the individual trajectories.

Reply: We agree with the Referee that the HYSPLIT model might not work properly when the arrival height was set as low as 150 m in this study. Hence, we performed the back trajectory analysis using the HYSPLIT model for arrival heights at 300 m, 700 m, and 1000 m above ground level. For each arrival height, cluster analysis was then used to group these entire 156 back trajectories into mean trajectories, e.g. clusters for different experimental periods of interest. The basic principle is to merge trajectories that are spatially near each other and form groups or clusters that represent these trajectories. This is computationally achieved by minimizing the differences between trajectories within a cluster and maximizing the differences between clusters. The detailed equations and calculation procedure are available in the HYSPLIT Tutorial document (https://www.ready.noaa.gov/documents/Tutorial/html/traj_clus.html). Results of the back trajectory analysis for different arrival heights are illustrated in Fig. R1, Fig. R2 and Fig. R3 in this response respectively. There are no significant differences between the results of the three different arrival heights. However, we agree with the referee that the arrival height of 150 m is fairly low for this model. Therefore, we switched to the arrival height of 700 m, which is the

mean height of the boundary layer in Guangzhou during the experimental period according to the data obtained from European Centre for Medium-Range Weather Forecasts (ERA Interim).

3. Aerosol particles are usually grouped as: nearly hydrophobic ($\kappa$<0.10), less hygroscopic ($\kappa$≈0.10-0.20) and more hygroscopic ($\kappa$>0.20; Liu et al., Atmos. Chem. Phys., 11, 3479– 3494, 2011). The authors may want to follow this attitude, or explain and argue for their own classification.

Reply: It is a conventional way to group aerosol particles into three characterized modes regarding their hygroscopicity. However, this might not always be the case; for instance, the other studies categorized aerosol particles into two modes, namely non or less-hygroscopic mode and more-hygroscopic mode (Aklilu et al., 2006; Gysel et al., 2007; Tan et al., 2013; Yeung et al., 2014; Wu et al., 2016) according to their own data. We tried to fit the measured data into three modes; however, the fitting procedure failed for more than half of the data sets. Two-mode fitting procedure was then carried out for all of the data.

4. The authors are requested to extend the MS with explicit discussions and estimations of the uncertainty of their major conclusions to prove their significance. In addition to that, smaller changes throughout the MS, e.g. showing standard deviations of slopes for correlation lines in Fig. 10, and similar amendments could also be adopted.

Reply: Yes, we have made a comprehensive uncertainty analysis regarding our measurements and calculations. Hence, the whole section (Sect. 3.4) was revised as shown below. **In addition, we feel the previous title might not fit for the current study and we changed it to '
[revised manuscript text omitted]

Minor comments:

1. It is disturbing that the terms aerosol – aerosols – aerosol particles are not used in a consistent manner in the MS (e.g. lines 64, 123 and 196). The authors should decide which option to use, and should adopt it in a coherent nomenclature.

Reply: I decided to use 'aerosol particles' in the revised manuscript to keep a coherent nomenclature.

2. Abbreviation PM stands for particulate matter (as correctly stated in line 191), but it is sometimes used for particle matter mass (e.g. line 30). The explanation of the abbreviation should be given at its first occurrence, and it is redundant to repeated it e.g. in lines 232– 233. Furthermore, simply write for instance: "The PM2.5 mass concentration varied" instead of "The range of particle mass concentration (PM2.5) varied". In addition, it is not the range that varies.

Reply: I revised the sentence to: 'The PM2.5 mass concentration varied from 20 to 180 ug/m$^3$, with relatively low values (roughly below 50 ug/m$^3$) during most of the time.'

3. Hygroscopicity usually refers only to sub-saturated conditions. Clarify line 65, or give appropriate references to back your statement.

Reply: References were added to back up my statement: 'Aerosol hygroscopicity describes the interaction between aerosol particles and ambient water molecules at both sub- and super-saturated conditions in the atmosphere (Topping et al., 2005; McFiggans et al., 2006; Swietlicki et al., 2008).'

4. Micron (e.g. lines 98, 177, Fig. 2) is not an SI unit. Micrometer should be used instead.

Reply: 'Submicron' was used as a definition for particles with sizes smaller than 1 micrometer in diameter. This term is widely used in aerosol research articles.

5. Remove the repetitions in lines 102–103 considering lines 67–69.

Reply: I modified the sentence to: 'Hygroscopicity, as an important physico-chemical property of atmospheric particles (Cheng et al., 2008; Gunthe et al., 2011; Cheng et al., 2016), has also been implemented into extensive campaigns in densely populated areas, such as North China Plain (Massling et al., 2009; Liu et al., 2011) and Yangtze River Delta (Ye et al., 2013).'

6. Revisit "self-assembly" (line 120) and "self-assembled" (line 152), or use perhaps laboratory made instead.

Reply: Yes, I used 'self-assembled' in the revised manuscript.

7. Clarify lines 185–189.

Reply: Wu et al. (2009) compared the BC concentration in PM1 and PM2.5, respectively, and found that BC aerosols mainly exist in the fine particles with roughly 80% of the BC mass in PM1. Due to the limited literature data on BC size distributions in the PRD region, we used this simplified assumption by Wu et al. (2009) to estimate the BC concentration in PM1 for this study.

8. Abscissa of Figs. 1 and 2 shows rather Date than Time, and its format of e.g. "(dd-MM)" could also be indicated. Furthermore, they could have (this) consistent format in both figures. In Fig. 2, how can be the probability in colour coding larger than 1? Explain or modify this. Put word space between measured values and their units everywhere in the figures. Extend the interpretation of your finding and discussions related to Fig. 2 within the frame of the conjunct conclusions of Cheung et al., Atmos. Chem. Phys., 16, 8431– 8446, 2016.

Reply: Figure 1 in the manuscript was modified and is shown below as Fig.R7. The color bar in Fig. 2 in the manuscript indicates the probability density rather than probability. To integrate the probability density along the x-axis, we will obtain the probability value. This is why its value is larger than one. All figures were modified as suggested.

9. Figures 3, 4 and 6 show the mean diurnal variations; the label of abscissa should be consistent in three figures; remove the tick label at 25; the time unit as "(HH)" should also be indicated; "during this study" is redundant; avoid abbreviations in the figure captions everywhere. Extend the interpretation of your finding and discussions related to the diurnal plots (Figs. 3 and 4 or Sects. 3.2 and 3.3) within the frame of the similar recent data of Enroth et al., Atmos. Chem. Phys. Discuss., https://doi.org/10.5194/acp-2017-926, 2017.

Reply: I moved Fig. 3 in the manuscript into supplement material part and modified all figures as suggested. Brief comparison was made between our results and Enroth et al. (2018).

10. All correlation scatter plots should have squared layout to facilitate better their interpretations.

Reply: Figures with x-axis and y-axis indicating the same variables was modified to have squared layout, see Fig. 4, Fig. 5 and Fig. 6.

11. Rounding off strategy should be revised throughout the MS; e.g. of $R^2$ in the figures or HGForg in lines 399, 405 ("1 and 1.3"), 515 or 546.

Reply: Yes, I modified all values I used in the whole manuscript to follow consistent strategy.

12. Lines 288–289: Remove "black carbon or" from "with black carbon or soot".

Reply: Yes, I removed 'black carbon or' from the text.

13. Line 319: Consider writing "compounds with different water uptake ability" instead of "compounds of different water uptake ability".

Reply: Yes, I changed it to 'compounds with different water uptake ability' in Line 319 in the revised manuscript.

[Figure]

Figure R1: The major clusters for the 72-hour backward trajectory simulation for air masses arriving at the CAWNET Panyu site at a height of 300 m. The upper panel shows the results throughout the whole observational period, while the lower panel on the left side shows the one during polluted days and the one on the right-hand side is for clean days.

[Figure]

Figure R2: The major clusters for the 72-hour backward trajectory simulation for air masses arriving at the CAWNET Panyu site at a height of 700 m. The upper panel shows the results throughout the whole observational period, while the lower panel on the left side shows the one during polluted days and the one on the right-hand side is for clean days.

[Figure]

Figure R3: The major clusters for the 72-hour backward trajectory simulation for air masses arriving at the CAWNET Panyu site at a height of 1000 m. The upper panel shows the results throughout the whole observational period, while the lower panel on the left side shows the one during polluted days and the one on the right-hand side is for clean days.

[Figure]

Figure R4: Closure study between the HTDMA-measured HGFs and the ACSM-derived HGFs. The black solid lines indicate the 1:1 line and the black dash lines represent ±10% deviation, while the red lines are the lines fitted to the data points. The color bar indicates the O:C ratio of the organic aerosol fraction.

[Figure]

Figure R5: Closure study between the HTDMA-measured HGFs and the ACSM-derived HGFs assuming the average inorganic mass fraction of PM1 were about 25%± 3% and 16% ±3% higher and the average ammonium sulfate mass fraction of PM1 were about 25%± 3% and 16% ±3% lower than those of 100 nm and 145 nm particles. The black solid lines indicate the 1:1 line and the black dash lines represent ±10% deviation, while the red lines are the lines fitted to the data points. The color bar indicates the O:C ratio of the organic aerosol fraction.

[Figure]

Figure R6: Closure analysis with the best fitting between the measured HGFs and the ACSM-derived ones using the O:C-dependent HGForg. The assumption of size-dependent chemical composition of aerosols was considered to determine the ACSM_derived HGF. The equation is the achieved approximation for HGForg as a function of the O:C of organic aerosol fraction.

[Figure]

Figure R7: Time series for relative humidity (RH), wind speeds, wind directions and concentrations of PM2.5 as well as BC concentration (bottom panel).

---

## Author Comment (AC2) · 9 Jul 2018

Answers to Referee #2

The authors appreciate the time the reviewer has spent on our manuscript. We thank a lot for the concern raised by the reviewer here, which we agree we did not notice and consider earlier. Hence, we tried our best to include more assumptions and uncertainty analysis into the revised version of the manuscript to address the comments and suggestions given by the reviewer. We hope the new version of the manuscript could meet the requirements for publication.

Major comments:

1. The closure analysis compares ACSM-derived HGF for PM1 with HTDMA-measured HGF for 100 nm or 150 nm particles. What was the mode diameter (in terms of mass diameter distribution) of the particle population of PM1? If the mode diameter of PM1 is significantly larger than 150 nm, the estimated HGForg can be significantly underestimated, because the hygroscopicity generally increases with the diameter in the accumulation mode (at least for 50 nm - 250 nm particles as shown in Liu et al. ACP, 2011). Is it possible that the estimated low HGForg is affected by such systematic biases?

2. In line 328-331, the authors mentioned that the measured HGF values for the mixed particles are less sensitive to the relationship of HGForg vs. O:C. I agree with this statement. However, this implies that the derived parameterization of HGForg = 0.3*O:C + 0.87 can be highly uncertain, as it is not well constrained by the measurements. I noticed that including the O:C-dependent HGForg only improves R^2 from 0.5 to 0.51 (Fig.10 vs. Fig. 9 d). In this case, what is the error associated with the parameterization?

3. Related to the previous comment, the proposed relationship would predict a HGForg value smaller than 1 for O:C < 0.43. This can be misleading. What does a HGForg value smaller than 1 mean? Is it just because of measurement errors?

4. The authors reported a best-fit HGForg value of 1.1. The error bar associated with this value should also be reported. Errors of both HTDMA and ACSM, along with the uncertainties associated bulk vs size-resolved closure analysis should be considered.

5. Even if the HGForg = 1.1 value is accurate, I don't think the authors can assert that "the CCN concentrations would probably be over-predicted as the organic material in these locations appeared to be close to hydrophobic". Several studies have shown that the CCN activity can be higher than that predicted based on the HGForg at _90% RH. For example, alpha-pinene-derived SOA has a kappa value of _0.04 at 90% RH, but the CCN-derived kappa value is around 0.1 (see Pajunoja et al., 2015), which is close to the value used in climate models. The low hygroscopicity derived at subsaturation regime does not necessarily indicate a low CCN activity.

Reply: As all of the above questions are related to each other, so we combine the answers to address the comments together.

Number size distributions of particles within the diameter range 10 - 400 nm was measured by the SMPS in this study. Hence, the mode diameter in terms of mass diameter distribution of the particle population of PM1 could not be determined directly. Cai et al. (2017) showed that the mode diameter of the mass size distribution of the chemical composition of PM1 was around 300 - 400nm for aerosols sampled at the same measurement site during the same season of 2014.

We agree that the estimated HGForg could be underestimated. However, size-resolved chemical composition information of ambient aerosols for current study is not available. We therefore considered the differences in the mass fraction of each component between PM1 and 145 nm particles obtained from Cai et al. (2017) and applied this assumption with certain uncertainties (20%) into the current study to make further evaluations. A newly determined parameterization of HGForg as a function of O:C ratio was given in the revised manuscript. We also performed a comprehensive uncertainty analysis for the hygroscopicity-composition closure and gave potential reasons from other sources of errors associated within this study.

With the new derived parameterization between HGForg and O:C, the values of HGForg values were all above one, which also indicates the size-resolved chemical composition taken from the above-mentioned assumption is physically reasonable. **The revised sections are attached below.**

We agree with the reviewer that the sentence 'the CCN concentrations would probably be over-predicted as the organic material in these locations appeared to be close to hydrophobic' is not properly used here. Hence, we deleted it.

**In conclusion, we feel the previous title might not fit for the current study and we changed it to 'Mixing state and particle hygroscopicity of organic-dominated aerosols over the Pearl River Delta Region in China'.**

Technical issues:

1. The authors should describe how the HTDMA was calibrated and how the data inversion was performed. Did the measurements reproduce literature values for pure compounds (e.g., ammonium sulfate)? Was the RH sensor calibrated? Did the two DMAs have a shift in diameter? Was the DMA transfer function considered in data inversion?

Reply: The calibration of the HTDMA was mainly performed in two steps. First, the electronic mobility diameter of particles was calibrated using polystyrene latex (PSL) spheres with known sizes. If the mean particle sizes measured by both DMAs are within the nominal sizes of PSL spheres, then both DMAs are capable to select a certain sized particles with required accuracy. Otherwise, the voltage supplying to either DMA should be corrected by using a voltage reference (here, as a voltage meter) until the nominal sizes of PSL is reached. Secondly, calibration of the HTDMA was done by measuring the HGF of ammonium sulfate at a certain RH, for instance, 90%. The measured HGF values for ammonium sulfate at a certain RH should match rather well with theoretical predictions (for instance, HGF=1.71 at RH=90%) with about 2% deviation. The measured growth factor distributions were fitted into bimodal lognormal distributions using a corrected data inversion approach by TDMAfit algorithm assuming all particles follow a Gaussian distribution (Stolzenburg, 1988; Stolzenburg and McMurry, 2008) The detailed description about calibration of the instrument and the data inversion approach for this study is given in Tan et al. (2013).

2. The O:C ratio was estimated from the f44 measured by a quadrupole ACSM, presumably less accurate than that measured by the HR-ToF-AMS. What was the uncertainty of the ACSM-derived O:C?

Reply: The uncertainties of the HR-ToF-AMS to quantify O:C have been discussed previously and are approximately to be ±0.1 as determined by Aiken et al. (2007) and Aiken et al. (2008). The uncertainties of determining O:C for ambient aerosols using other types of AMS and

ACSM is not well established. However, Ng et al. (2011) demonstrated that due to the similarities with AMS, all the methods that already have been developed for the usage of AMS are capable to be used in ACSM data analysis. The largest differences between ACMS and AMS are that ACSM is not designed to measure mass size distribution and it has a lower detection limit (e.g. 0.2 ug/m3). Hence, we believe that the O:C ratio determined by ACSM should not deviate significantly than the one from AMS.

3. Figure 2. Unit of the color bar is missing.

Reply: We added the unit for the color bar in Fig. 2 in the manuscript.

The revised section is attached as below:

Section 3.4

[revised manuscript text omitted]
 by taking into account of size-dependent chemical composition of aerosols during polluted and clean days. The black solid lines indicate the 1:1 line and the black dash lines represent ±10% deviation, while the red lines are the lines fitted to the data points. The color bar indicates the O:C ratio of the organic aerosol fraction.

[Figure]

Figure R5: Closure analysis with the best fitting between the measured HGFs and the ACSM-derived ones using the O:C-dependent HGForg during polluted and clean days. The assumption of size-dependent chemical composition of aerosols was considered to determine the ACSM_derived HGF. The equation is the achieved approximation for HGForg as a function of the O:C of organic aerosol fraction.

[Figure]

Figure R6: Comparison with earlier studies on the hygroscopicity of organic material with atomic O:C ratio (or *f44* from chemical composition data) obtained from different environmental background areas. Other studies were using derived κorg, while this study is using HGForg for the hygroscopicity of organic material.

[revised manuscript text omitted]

---

## Author Comment (AC3) · 9 Jul 2018

Answers to Referee #3

The authors appreciate the time the reviewer has spent on our manuscript, assisting us to produce a higher quality, understandable publication. All the requested comments and suggestions are addressed and introduced to the revised version of the manuscript.

The authors present a study of low hygroscopicity of organic material in anthropogenic aerosols under pollution episode in China. The data set is rich, but the manuscript has two major deficiencies that should be addressed prior to considering further review. (1) The upper particle size ranges detected by different instruments are quite different: 145 nm for HTDMA, 1um for ACSM and 2.5 um for Aethalometer. Before drawing any conclusions, the authors should consider the uncertainties caused by different size ranges of particles being measured when integrating all dataset. (2) Most of previous studies showed that the hygroscopic growth factors of secondary organic aerosols are below 1.2. For example, the water-soluble organic carbon measured by Martin Gysel et al., has a hygroscopic growth factor of up to 1.17. In this study, the HGForg below 1.1 is actually comparable to those in other observations. Therefore, the low hygroscopicity of organic material cannot be a compelling result. Reference Gysel, M., et al., Hygroscopic properties of water-soluble matter and humic-like organics in atmospheric fine aerosol. Atmos. Chem. Phys., 2004. 4(1): p. 35-50.

Reply: Yes, the Reviewer is right that the upper particle size ranges detected by different instruments are quite different. For BC measurements, Wu et al. (2009) compared the BC concentration in PM1 and PM2.5, respectively and found that BC aerosols mainly exist in the fine particles with roughly 80% of the BC mass in PM1. Due to the limited literature data on BC size distributions in the PRD region, we used this simplified assumption by Wu et al. (2009) to estimate the BC concentration in PM1 for this study. In addition, the ACSM-derived HGF is not sensitive to the change in BC mass concentration, which was also discussed in the uncertainty analysis in the revised manuscript; see the attached Section (text in blue).

For the other chemical components measurements, ACSM only measured their mass concentration in PM1, which should not be compared directly with the one measured by HTDMA with a certain size. However, size-resolved chemical composition information of ambient aerosols for current study is not available. Cai et al. (2017) measured size-resolved chemical composition of ambient aerosols obtained from the same measurement site during the same season of 2014. Their results showed that the average organic mass fraction of PM1 were about 16% lower than that of 145 nm particles. We therefore considered the differences of mass fraction of each component in between PM1 and 145 nm particles obtained from Cai et al. (2017) and applied this assumption with certain uncertainties into the current study to make further evaluations. A newly determined parameterization of HGForg as a function of O:C ratio was given in the revised manuscript. We also performed a comprehensive uncertainty analysis for the hygroscopicity-composition closure and gave potential reasons from other sources of errors associated within this study.

For the second comment: After taking into account of the size-dependent chemical composition of aerosols, HGForg value was shifted from 1.1 to 1.26, which was close to the one as 1.18 used by Yeung et al. (2014) measuring the hygroscopicity of ambient aerosols in September 2011 at the HKUST Supersite, less than 120 km away from our measurement site. I agree with the reviewer that the determined hygroscopicity of organic material is not significantly different from the ones from other

studies. However, without identifying the hygroscopicity of organic material for current environmental background, one should always use the assumed parametrization for the hygroscopicity of organic material from other environments with big cautions. In addition, this study is, to our knowledge, the first time to identify the hygroscopic properties of the organic material and their O:C dependency, which may help us understand the chemical composition, sources and aging processes of atmospheric aerosols in this region. **After considering the difference of chemical composition between size-resolved aerosols and bulk ones, we feel the previous title might not fit for the current study and we changed it to '
[revised manuscript text omitted]

---

## Author Comment (AC4) · 9 Jul 2018

Answers to Referee #4

The authors appreciate the time the reviewer has spent on our manuscript, assisting us to produce a higher quality, understandable publication. The requested comments and suggestions are addressed and introduced to the revised version of the manuscript.

General Comments:

This paper is attempting to compare many components of the measured aerosol: four different aerosol sizes, two different HGF modes, clean vs polluted conditions, and diurnal trends. It's a complicated set of comparisons, and different sections of the paper address different things. The reader would benefit if the authors would more clearly state what each section is comparing, and only include the most relevant comparisons. For example, the first paragraph of section 3.2 discusses the diurnal trends of the mean HGF, but quickly states that there are no significant trends – probably because in the next section we can see that the LH and MH modes have opposite trends, and the mean HGF, which is the average of the two, sees these trends cancel out. So why include the mean HGF at all? Another example is at the end of section 3.3, where the authors demonstrate that the HGF dependence is different in clean and polluted conditions. If this is true, the authors should be careful in the rest of the paper to distinguish between clean and polluted conditions in their other analyses.

Reply: We shortened the paragraph discussing the diurnal variation of the mean HGF and restructured the whole section substantially as requested. We also made closure analysis for polluted and clean days similar as Sect. 3.4 to improve the Results and Discussion section with more detailed arguments. The new section was attached below in blue.

[revised manuscript text omitted]
 by taking into account of size-dependent chemical composition of aerosols during polluted and clean days. The black solid lines indicate the 1:1 line and the black dash lines represent ±10% deviation, while the red lines are the lines fitted to the data points. The color bar indicates the O:C ratio of the organic aerosol fraction.

[Figure]

Figure R2: Closure analysis with the best fitting between the measured HGFs and the ACSM-derived ones using the O:C-dependent HGForg during polluted and clean days. The assumption of size-dependent chemical composition of aerosols was considered to determine the ACSM_derived HGF. The equation is the achieved approximation for HGForg as a function of the O:C of organic aerosol fraction.

[Figure]

Figure R3: Comparison with earlier studies on the hygroscopicity of organic material with atomic O:C ratio (or *f44* from chemical composition data) obtained from different environmental background areas. Other studies were using derived κorg, while this study is using HGForg for the hygroscopicity of organic material.

Secondly, the authors should identify the primary message of the manuscript and more clearly describe this result. Is it that the hygroscopicity's lower-than-expected dependence on O:C is attributed to a higher concentration of organics with larger molecular weights? If so, the authors should discuss this further. Are there experimental measurements available to support this? If this is the main conclusion, what should the reader learn from the extensive look into the dependence on inorganics, on the diurnal averages, which is what the majority of the results section is about?

Reply: According to the other Referees' comments, ACSM-derived HGF should not be directly compared with the one measured by HTDMA, since ACSM measured the bulk aerosols while HTDMA measured the HGF of size-resolved particles. We therefore considered the differences of mass fraction of each component between PM1 and 145 nm particles obtained from Cai et al. (2017), which measured the size-resolved chemical composition of ambient aerosols obtained from the same measurement site during the same season of 2014 and applied this assumption with certain uncertainties into current study to make further evaluations. A newly determined parameterization of HGForg and its O:C dependency was given in the revised manuscript. We also performed a comprehensive uncertainty analysis for the hygroscopicity-composition closure and gave potential reasons from other sources of errors associated within this study. After taking into account of the size-dependent chemical composition of aerosols, HGForg value was shifted from 1.10 to 1.26. We hence feel our previous discussion and arguments were improper given. Hence we deleted related arguments at several places and modified the body text. However, even considering the size-resolved chemical composition of aerosols, we still observed the hygrosocpicity of organic compounds in current study has a relatively low O:C dependence. This finding is not completely unexpected, as Wu et al. (2016), which studied the particle hygroscopicity in the urban atmosphere of Beijing, also observed similar relation between HGForg and O:C ratio of ambient aerosols. Wu et al. (2016) discussed that the addition of both alcohol and carboxylic functions could elevate the O:C ratio of organic aerosols but form species with different hygroscopicities. This could be a possible reason to explain that the variation of O:C of organic aerosols is not necessarily responsible for the changes in hygroscopicity. This comparison and possible discussions were given in the revised manuscript.

Specific Issues:

Line 86 - 90: What is the relevant difference here between oxidation level and the oxygenation state? Insert a sentence here detailing why oxidation level is theoretically correlated with water uptake, since this is an important part of the results.

Reply: The oxygenation state is a more robust measure of the degree of oxidation for organic aerosols, while oxidation level, represented as O:C, sometimes may be affected by hydration and dehydration processes taking place in the atmosphere. We added a sentence into the text: They found that the oxidation level or the oxygenation state of the entire organics, which directly affects their corresponding solubility in water, is the major factor drives the water uptake ability of the organic fraction in aerosols.

Line 90 - 92: It is stated that the knowledge of the dependency of hygroscopicity on oxidation level is unknown in urban China. Since this is the main focus of the paper, include a line indicating why this environment is different.

Reply: However, knowledge on the hygroscopicity of organic material and its dependency on the oxidation level of organics in urban background areas under high aerosol mass loading

conditions, for instance, in China, where air pollution has become one of the top environmental concerns (Chan et al., 2008), is limited.

Line 152 - 173: More details about the HTDMA should be included in this section. The second DMA is operating in SMPS mode? How fast/frequent are the scans and therefore what is the time-resolution for retrieval of the HGF? How frequently does the first DMA cycle between the 4 diameter set points? How are doubly- and triplycharged particles that are transmitted by the first DMA handled? Are the particle size distributions plotted in the bottom frame of Figure 2 from SMPS scans by the first DMA or from some other technique?

Reply: Before operating in HTDMA mode, the second DMA was bypassed. The aerosol particles, after being introduced into the first DMA, was directed into the CPC to measure the number size distribution of the ambient aerosols. This is how we obtained the SMPS scans. Hence, the bottom frame of Fig. 2 in the manuscript is directly from our HTDMA system when SMPS mode is on. The time resolution for a whole scan including the SMPS scans and HTDMA scans of particles of 4 different sizes are around one hour. Multiply charge correction was performed during data inversion procedure according to the method introduced by Gysel et al. (2009).

Line 178: What are 'Ambient-improved' ratios? Either define this term or leave it out and direct the reader to a reference.

Reply: We changed the sentence to: 'The oxygen to carbon (O:C) were then estimated by their relationship to the mass fractions of m/z44 (f44) to the total organic mass according to Canagaratna et al. (2015).'

Line 184 - 186: Briefly state what the simplified approach is. Is all the BC assumed to be in PM1? Or a weighted fraction?

Reply: Wu et al. (2009) compared the BC concentration in PM1 and PM2.5, respectively, and found that BC aerosols mainly exist in the fine particles with roughly 80% of the BC mass in PM1. Due to the limited literature data on BC size distributions in the PRD region, we used this simplified assumption by Wu et al. (2009) to estimate the BC concentration in PM1 for this study.

Line 188: The line "individual size bins" is confusing. I assume the authors are referring to the 4 sizes selected by the first DMA? Replace with something similar to "the ACSM measures only accumulation mode aerosol, and therefore the Aitken mode particles may have a different chemical composition".

Reply: We changed the sentence to: 'It is necessary to note that the ACSM measures the chemical composition of bulk aerosols, which may be significantly different from those of Aitken mode particles.'

Line 191: Briefly state what instrument was measuring the PM2.5 chemical concentrations. An AMS?

Reply: The PM2.5 mass concentration was measured by an Environmental Dust Monitor (EDM, Grimm model 180).

Line 209 - 212: What is the justification for assuming the aerosol is completely neutralized? What would the effect be on the results be if it were not completely neutralized?

Reply: We are sorry that this assumption is based on previous studies (Gysel et al., 2007), which used similar ion paring scheme, and we do not fully understand the effect if the aerosols are not completely neutralized. However, we hypothesize that if our aerosols are more acid, the HGF would be higher than the predicted one as inorganic acid is more hygroscopic than its neutral form.

Line 272 - 282: See comment in General Comments. The paragraph is perhaps un necessary. What can be learned from looking at the diurnal profile of the mean HGF that isn't learned from looking at the MH and LH components separately?

Reply: We deleted this paragraph as suggested.

Line 296 - 299: What is the justification for the assertion that the MH mode particle experience a decrease in HGF during the day because they are uptaking less photoreactive species. Do typical reaction rates or back-of-the-envelope calculations support this assertion? Which species are involved? If this is true, how do the authors reconcile the fact the O:C ratio sharply increases during the day, and this paper indicates that there is at least a somewhat positive correlation between O:C and HGF?

Reply: I did not say they uptake less photoreactive species but rather organics from stronger photoreaction during daytime. These newly formed organics are expected to be less hygroscopic than those aged organics, which were already present in the aerosols. Here, these arguments are only related to MH mode particles, however, taking into account the mean HGF of the whole aerosol population as Fig.3 in the manuscipt, HGF values were actually higher during daytime, which is consistent with the increase in O:C during daytime.

Line 305: The authors state here that Hong 2015 and Cai 2017 report that the boundary layer height has an effect on aerosol populations, but later on line 378, they suggest it doesn't. This disagreement should be addressed more fully.

Reply: We deleted the sentence used in Line 305, but kept the one in line 378, since the logic in line 378 is more reasonable.

Line 323: The authors state that they can only compare HGFs from the HTDMA and ACSM for larger particles. But they have also demonstrated that larger and smaller particles behave differently. The authors should address any hypotheses for how HTDMA and ACSM might agree for smaller particles.

Reply: Size-dependent chemical composition should be considered into the derivation of HGF by ACSM data, thus the HTDMA_measured HGF then could be compared with the one from ACSM.

Line 325: State why HGF is expected to positively correlate with the inorganics/(organics + BC) ratio.

Reply: Inorganics are commonly quite hygroscopic, while relative to inorganics, organic and black carbon are less hygroscopic or non-hygroscopic. Hence, the more inorganics in the

aerosol phase, the larger HGF and the more organics or BC in the aerosols, HGF is relatively smaller.

Line 349 and 352: The authors state the percentages 64% and 21% in reference to the back trajectories without discussing where these numbers come from. Furthermore, more information about the trajectories would be helpful, such as error bars on those percentages.

Reply: The method we used here was called cluster analysis. The details of this method could be found in http://ready.arl.noaa.gov/HYSPLIT.php. Thus only a brief introduction would be given here. The trajectories that are near each other are merged to a mean trajectory to represent those groups. When the clustering is complete, the change in the Total Spatial Variance (TSV) as the trajectories is merged into one cluster. The TSV is computed from the position vectors for the individual trajectory and its cluster mean trajectory. According to the change of TSV, we decide the number of cluster as an appropriate solution (In this work, the number is 5). The percentage of the cluster means that how many back trajectories can be represent by this cluster. According to the principle of cluster analysis and the limit of this model, the standard deviation of those percentages is not available from the model calculation directly.

Line 354: Is there an observed increase in ACSM organics on days when the trajectories indicate air masses are arriving from the inland areas? If not, why is that?

Reply: This is not clearly seen in current study, as the meteorological conditions also influence the concentration of organics in aerosols, for instance, the stagnant weather conditions during Sep 22 to 27 with low wind speed favor the accumulation of atmospheric pollutants including the mass concentration of different species. However, the air masses were mainly from coastal areas.

Line 390: Do the authors have a suggestion for why this trend (HGF depends on O:C more during clean days than polluted days) is observed? It seems like an important result, yet isn't discussed extensively in the conclusions. Additionally, why is the parameterization of the HGF-to-O:C relationship not done separately for clear vs polluted days?

Reply: As previously stated in the manuscript, during polluted days, the aerosols appeared to be from long-range transported, having longer aging history. The organic material in these aerosols were fully oxygenated, even with various O:C ratio. However, during clean days, the aerosols was mainly from local emissions or formed locally without complex histories. The changes in HGForg revealed the oxidation state of theses locally formed organic material. The closure analysis to determine the parameterization of the HGF-to-O:C relationship was done separately for clean and polluted days as above.

Line 401: Is there an operational definition for suburban aerosol? Does this just mean an aerosol population that is somewhere between typical urban and rural characteristics?

Reply: Yes, compared to urban area, it has a lower density area that separate residential and commercial areas from one another. It could be part of the urban area or exist as a separate residential community within commuting distance of a city.

Line 402 - 405: More detail about the residual fit should be added here. Is the ZSR prediction compared to all the HTDMA measured HGF? Of all sizes? Or just the polluted or clear days? Are different values derived depending on the subset of measured data to compare to?

Reply: The ZSR prediction is compared with all the HTDMA-measured HGF. Moreover, this is also done for the polluted and clean days. This is included in the revised manuscript.

Line 415: Why was the ACSM not measuring size-selected aerosol in this study, as was done in Yeung et al?

Reply: ACSM is an instrument that cannot measure the size-resovled chemical compositon of aerosols, while Yeung et al. (2014) used a High Resolution Time-of-Flight Aerosol Mass Spectrometer, which is capable to obtain the size-segregated chemical composition information. Unfortunately, during our experimental period, we did not have HR-ToF-AMS.

Line 425: More information should be included about how this parameterization was derived. What parameters were allowed to vary, and what was the parameter that was minimized? Is a $R2$ of 0.51 significantly better than 0.5? In the next paragraph, an improved parameterization is introduced by allowing SOA density to vary. Which parameterization is better? Why does the conclusion section only mention this first parameterization?

Reply: We gave a revised section illustrating the closure analysis, which should be much easier to understand and more reasonable. In addition, we did not include other parametrization in the conclusion, but did a comprehensive uncertainty analysis to discuss the potential errors associated within the closure analysis.

Line 430: What is the justification for stating that the hygroscopicity of organics isn't affected by the presence of inorganics?

Reply: As the results shown, HGForg was actually quite constant during the whole study, while the concentration of inorganics varied. This is an indicator that the hygroscopicity of organics in our study is not observed to be affected by the presence of inorganics.

Line 444: How are the authors accounting for error here? Presumably there is error in the measurement, which propagates through to the derivation of the parameterization.

Reply: We performed a comprehensive uncertainty analysis for the hygroscopicity-composition closure and gave potential reasons from other sources of errors associated within this study, which is shown in the revised manuscript.

Line 490: Have the authors plotted the HGF vs the concentration of certain inorganics? Say, vs ammonium sulfate or sulfuric acid to see if there is a larger trend for compounds known to be more hygroscopic?

Reply: This is actually illustrated in Fig. 5 and Fig. 8. However, to plot the HGF vs. the concentration of ammonium sulfate may not indicate the complexity of ambient aerosols, as there was also substantial amount of organic species, which is less hygroscopic than ammonium sulfate. Even with increasing amount of ammonium sulfate concentration in aerosols, the increase of organics will cancel out the effect on HGF. This is the reason to plot HGF vs. Inorganics/(Organics+BC) is more reasonable.

Figure 2: Remove the dates from under each frame and just put them under the bottom frame. Color bar for the top four frames should be labeled. Additionally, it seems as though the MH and LH modes both have diurnal cycles between <1 and > 4. If this is simply because the total

number of particles has a diurnal profile, it would be easier to see this if it was normalized to the total number of particles. In the bottom plot, because there is only one point on the y-axis, it's hard to see that it's in log space. The boundaries (i.e. 10 -1000 nm) should be indicated, with ticks to show that it is logarithmic.

Reply: I modified the figure as suggested.

Figure 3: See Comment on line 272. It's possible that this figure is not needed.

Reply: I deleted this figure and the corresponding text in the revised manuscript.

Figure 4: Is this separately out for polluted or clear days? Why not?

Reply: We put this figure in the supplement.

Figure 5: What happens if these plots are made with MH or LH HGF instead of the mean?

Reply: We understood the reason why the referee asked us to do this. However, the particle phase chemical composition data as well as the O:C ratio is the mean value of the bulk aerosols. Information of the chemical composition of each mode is not available. Poor correlation was observed when plotting the MH and LH HGF with the bulk aerosol composition.

Figure 7: The colors for these trajectories should be labeled more clearly, and described more fully in the caption and also in the manuscript. Do they represent one representative trajectory? Or a weighted average? What was the spread on those individual trajectories?

Reply: These trajectories are the clusters, which represent a group of back trajectories. The meaning of cluster has been answered in the former comments.

Grammatical/Minor:

Line 102: What does "purposes" mean here? Do you mean "properties"?

Reply: I changed to 'properties'.

Line 107: PRD, not RPD

Reply: Yes, thanks for pointing it out and we revised it as suggested.

Line 155: Tan et al. 2013b doesn't appear to be in the listed references. Neither is Tan 2013a

Reply: I added them into the reference list in the revised manuscript.

Line 159: Why denote the dry mobility diameter as D0? Why not "Dp (0% RH)"?
Reply: It is a conventional way to denote the dry mobility diameter as D0, hence, I keep it as it is.

References:

[revised manuscript text omitted]

---

## Author Comment (AC5) · 9 Jul 2018

Answers to Referee #5

The authors appreciate the time the reviewer has spent on our manuscript, assisting us to produce a higher quality, understandable publication. The requested comments and suggestions are addressed and introduced to the revised version of the manuscript.

General Comments:

This manuscript presents results of simultaneous measurements of aerosol hygroscopicity and chemical composition in suburban site in Southern China. The measurement period covers almost 1 month and both polluted and relatively clean conditions were observed. I admit that such measurement can be very costly and labor intensive, and therefore the comprehensive set of data presented in the manuscript may carry certain value for the scientific community. However, with the current state of the manuscript, authors' main conclusion is very difficult to sink in for the readers. Authors seem to persist on determining the HGForg and large part of the manuscript is dedicated for that. However, in my opinion, it is obvious from the results that the oxidation level of organics does not affect the hygroscopicity of the suburban aerosols very much, and that might pretty much be the end of the story for HGFrog. Instead, I would like to see much more in-depth discussion on diurnal variations of LH mode in smaller particles (30 and 60 nm) and how the new particle formation and subsequent growth affects the aerosol hygroscopicity. I therefore recommend that the manuscript may be acceptable for publication in ACP after major restructuring.

Reply: We thank the referee's comments regarding the results of our manuscript. The referee is correct that the new particle formation and subsequent growth affects the aerosol hygroscopicity. At the referee's request, we did a case study of particle hygroscopicity during an NPF event.

Figure R1 shows the time series of particle number concentration, GF-PDFs of 30 and 145 nm particles and the bulk aerosol chemical composition of PM1 during 04. October, 2016, during which an NPF event was observed. The NPF event started at around 10:00 am, after which we observed a substantial increase in the number fraction of MH mode particles from 0.5 to around 1, for 30 and 145 nm particles. This indicates a clear conversion of particles from externally to internally mixing. Number fraction of the MH mode for 30 and 145 nm particles decreased again to 0.5 around 17:00 pm, which might be explained by the traffic emissions during the rush hours of the day. Meanwhile, HGF of MH mode of both 30 and 145 nm particles showed a slight decrease after the NPF event started. This suggests that the candidate for the material in the MH mode particles during NPF event may be not only sulfuric acid or ammonium sulfate but also secondary organic species. The contribution from secondary organic species, which are less hygroscopic than ammonium sulfate, may dominate in the newly formed particles to be able to reduce the HGF of pre-existing particles. Similar observations was found in other studies (Levin et al., 2012; Wu et al., 2015).

Hence, we believe the effect of NPF on hygroscopicity might be worth a study of its own. However, without size-resolved chemical composition of particles, we cannot make further conclusion based on current results. The findings above seem to be a little bit too weak to have a single section in the manuscript. To include these into this paper would make the manuscript unnecessarily long, which would even weaken the key points of this paper. At the other four

referees' requests, chemical composition of PM1 should not be compared with the one from size-segregated aerosols and uncertainty analysis should be implemented into the manuscript. Hence, we adopted some assumptions and performed a comprehensive uncertainty analysis regarding the hygroscopicity-composition closure. In addition, similar analysis was done for the polluted and clean days. Finally, we revised our conclusion substantially. The revised sections are attached after the answers of all comments with color in blue.

[Figure]

Figure R1: Time series of particle number size distribution, GF-PDFs of 30 and 145 nm particles and chemical composition of PM1 during 04, October 2016.

Specific comments

152: It is not clear from the manuscript how the HTDMA was operated to obtain the particle number size distribution (10-1000 nm) simultaneously while the instrument was measuring HGF in 4 size classes.

Reply: Before operating in HTDMA mode, the second DMA was bypassed. The aerosol particles, after being introduced into the first DMA, was directed into the CPC to measure the number size distribution of the ambient aerosols. This is how we obtained the SMPS scans. Hence, the bottom frame of Fig. 2 in the manuscript is directly from our HTDMA system when SMPS mode is on. We made a mistake before in the manuscript; actually particle number size distribution within 10-400 nm was measured, but not 10-1000nm particles. We corrected it in the revised manuscript.

175-178: It is critical to indicate the calibration procedure of ACSM and what calibration parameters were used (e.g. relative ionization efficiency of SO4). Such calibration parameters can critically affect the inorganic and organic mass fractions (and therefore the ensemble HGForg of 1.1).

Reply: Ammonium nitrate (AN) particles, generated by an atomizer, was selected by a DMA with a certain size at 300 nm. The size-selected particles were then introduced into a CPC and the ACSM. After obtaining the number concentration of AN particles, we calculated their mass concentration by multiplying the density of AN. The mass concentration obtained from the CPC was then compared with the one from ACSM. Then the corresponding RIE value for NH4 was calculated. Similar procedure was performed for ammonium sulfate particles. The RIE values for NH4 and SO4 were obtained as 5.63 and 0.78, respectively. The detailed calibration procedure was described in Ng et al. (2011).

300-303: The logical basis to support the following conclusion is not clear. "In case of smaller particles (30 nm, 60 nm), HGFs of MH group particles appeared to decrease during the afternoon until about 8:00 pm, suggesting that these particles were not long-range transported, but rather secondary formed either locally or from nearby emissions."

Reply: We changed the sentence to: 'In case of smaller particles (30 nm, 60 nm), HGFs of MH group particles appeared to decrease during the afternoon until about 8:00 pm. This is probably attributed to the intensive traffic emissions at the time of rush hour.'

462-519: Extra caution must be taken when comparing k based on supersaturation conditions and HGF based on sub-saturated conditions. The k derived from sub- and supersaturated conditions can be quite different in some cases. In such case, the discussion on potential bias on CCN concentration may not be relevant.

Reply: The last figure in the manuscript is only an illustration for the relation between the hygroscopicity of organic material and its O:C ratio. Kappa values are not directly compared with the ones of HGF. We agree with the referee that kappa derived from sub- and supersaturated conditions can be quite different in some cases. Hence, we deleted the discussion on the calculation of CCN concentration. New discussions were included in the revised manuscript.

Technical corrections

120: "self-assembly" should appear "self-assembled"

Reply: We rephrased it to 'self-assembled'.

188: what does it mean by "individual size bins"?

Reply: We changed it to: 'It is necessary to note that the chemical composition of PM1 can be different from those of size-selected aerosol particles'.

330: rephrase "uncertainties of in growth factor"

Reply: We changed it to 'uncertainties in growth factor'.

423: "as followed" should appear "as follows"

Reply: We changed it to 'as follows'.

Section 3.4

[revised manuscript text omitted]
 by taking into account of size-dependent chemical composition of aerosols during polluted and clean days. The black solid lines indicate the 1:1 line and the black dash lines represent ±10% deviation, while the red lines are the lines fitted to the data points. The color bar indicates the O:C ratio of the organic aerosol fraction.

[Figure]

Figure R6: Closure analysis with the best fitting between the measured HGFs and the ACSM-derived ones using the O:C-dependent HGForg during polluted and clean days. The assumption of size-dependent chemical composition of aerosols was considered to determine the ACSM_derived HGF. The equation is the achieved approximation for HGForg as a function of the O:C of organic aerosol fraction.

[Figure]

Figure R7: Comparison with earlier studies on the hygroscopicity of organic material with atomic O:C ratio (or *f44* from chemical composition data) obtained from different environmental background areas. Other studies were using derived κorg, while this study is using HGForg for the hygroscopicity of organic material.

[revised manuscript text omitted]

---

## Referee Report (RR1)

This draft of the paper is much improved. I would recommend publication, following some minor revisions.

Section 3.2, last paragraph – The authors state that there is good correlation between the HGF vs Inorg/(Org + BC) ($r2 = 0.38 – 0.47$), but not good correlation between HGF and O:C ($r2 = 0.23$). These r2 values do not seem so different to me as to state that one shows good correlation and the other bad. The authors should change the phrasing there to simply say that one has better correlation than the other. Unless, of course, there is some other justification for this ranking that I'm not aware of. In that case, state the justification.

Section 3.4.2 – Switch this section with section 3.4.3 for a more natural flow.

Section 3.5 – Why is this called "Synthetic Comparisons". I would call it something like "Comparison to other ambient measurements", because only one of the studies you compare it to is a lab-based study.

Figure 1 – Put a marker on the x-axis that indicates which time periods are considered polluted vs. clean. This will make it visually more clear.

Figure 5 – Y-axis labels need units, and the 4 plots are too close together.

Figure 6 – Clearly label the three plots as polluted, clear, etc. Also, it would be helpful if all three were on the same latitude and longitude scale.

Figure 12 – Mei et al is misspelled as "Mie" in the plot legend.

All figures – standardize the formatting for all figures.

---

## Author Response (AR2)

We appreciate these practical suggestions from the reviewer. These technical corrections are essentials to improve the quality of our manuscript. Please find included a point-by-point response to all the comments.

Answers to Referee

Section 3.2, last paragraph – The authors state that there is good correlation between the HGF vs Inorg/(Org + BC) (r2 = 0.38 – 0.47), but not good correlation between HGF and O:C (r2 = 0.23). These r2 values do not seem so different to me as to state that one shows good correlation and the other bad. The authors should change the phrasing there to simply say that one has better correlation than the other. Unless, of course, there is some other justification for this ranking that I'm not aware of. In that case, state the justification.

Reply: The sentence was rephrased as 'In Fig. 4, the HGFs of accumulation mode particles correlate with the mass fraction ratio between inorganics and organics + BC ($R^2$ = 0.38 - 0.47) better than the oxidation level of the organic fraction with $R^2$ values of around 0.23.'

Section 3.4.2 – Switch this section with section 3.4.3 for a more natural flow.

Reply: We switched these two section with each other.

Section 3.5 – Why is this called "Synthetic Comparisons". I would call it something like "Comparison to other ambient measurements", because only one of the studies you compare it to is a lab-based study.

Reply: We changed it to 'Comparison to other ambient measurements'.

Figure 1 – Put a marker on the x-axis that indicates which time periods are considered polluted vs. clean. This will make it visually more clear.

Reply: We indicated these two periods in Fig. 1 in the revised manuscript.

Figure 5 – Y-axis labels need units, and the 4 plots are too close together.

Reply: Yes, we modified this figure as suggested.

Figure 6 – Clearly label the three plots as polluted, clear, etc. Also, it would be helpful if all three were on the same latitude and longitude scale.

Reply: We added these labels in Fig.6 in the revised manuscript. However, the resolution of the figure will be reduced a lot if we put these subplots on the same latitude and longitude scale. Hence, we kept the previous version.

Figure 12 – Mei et al is misspelled as "Mie" in the plot legend.

Reply: We corrected it.

All figures – standardize the formatting for all figures

Reply: We replot all of the figures to have similar formats.